# MMCOMPOSITION: REVISITING THE COMPOSITIONALITY OF PRE-TRAINED VISION-LANGUAGE MODELS

## ABSTRACT

The advent of large Vision-Language Models (VLMs) has significantly advanced multimodal understanding, enabling more sophisticated and accurate integration of visual and textual information across various tasks, including image and video captioning, visual question answering, and cross-modal retrieval. Despite VLMs' superior capabilities, researchers lack a comprehensive understanding of their compositionality – the ability to understand and produce novel combinations of known visual and textual components. Prior benchmarks provide only a relatively rough compositionality evaluation from the perspectives of objects, relations, and attributes while neglecting deeper reasoning about object interactions, counting, and complex compositions. However, compositionality is a critical ability that facilitates coherent reasoning and understanding across modalities for VLMs. To address this limitation, we propose **MMCOMPOSITION**, a novel human-annotated benchmark for comprehensively and accurately evaluating VLMs' compositionality. Our proposed benchmark serves as a complement to these earlier works. With MMCOMPOSITION, we can quantify and explore the compositionality of the mainstream VLMs. Surprisingly, we find GPT-4o's compositionality inferior to the best open-source model, and we analyze the underlying reasons. Our experimental analysis reveals the limitations of VLMs in fine-grained compositional perception and reasoning, and points to areas for improvement in VLM design and training. [1]

## 1 INTRODUCTION

Pre-trained vision-language models, such as GPT-4o (Achiam et al., 2023), LLaVA (Liu et al., 2024b), InternVL (Chen et al., 2024b), and VILA (Lin et al., 2024), have demonstrated impressive capabilities in complex reasoning, and have achieved remarkable results in various vision-language (VL) tasks. Despite these advancements, contemporary state-of-the-art VLMs still struggle with understanding fine-grained multimodal compositional information (Yuksekgonul et al., 2022; Thrush et al., 2022). For instance, VLMs often fail at counting objects in images, especially when the objects are mixed with other items or occluded, while humans can handle this task easily. This reveals a compositionality gap between humans and models. However, *compositionality* is recognized as a core capability for VLMs (Yuksekgonul et al., 2022), referring to the ability to understand and produce a potentially infinite number of novel combinations of known visual and textual components, i.e., to make "infinite use of finite means" (Chomsky, 2014). Compositionality is essential for tackling challenging questions in image captioning, visual question answering (VQA), and scene understanding, where complex interactions between objects and attributes need to be communicated in natural language.

In recent years, there has been a growing focus on evaluating the comprehensive capabilities of large VL models, such as MMBench (Liu et al., 2023b), MMMU (Yue et al., 2023), MMVet (Yu et al., 2024a;b), MME (Fu et al., 2023), Seed-bench (Li et al., 2023a), MMStar (Chen et al., 2024a), MathVista (Lu et al., 2023), and LLaVA-Bench (Liu et al., 2024b). These benchmarks evaluate VLMs' capabilities in recognition, OCR, knowledge, language generation, spatial awareness, and mathematical reasoning. While some of these benchmarks include visual compositional question-answering (QA) pairs (Fu et al., 2024; Li et al., 2023a; Tong et al., 2024b), none are specifically designed to comprehensively evaluate the models' fine-grained VL compositional perception and

---

[1]All data and code will be released upon publication of this paper.

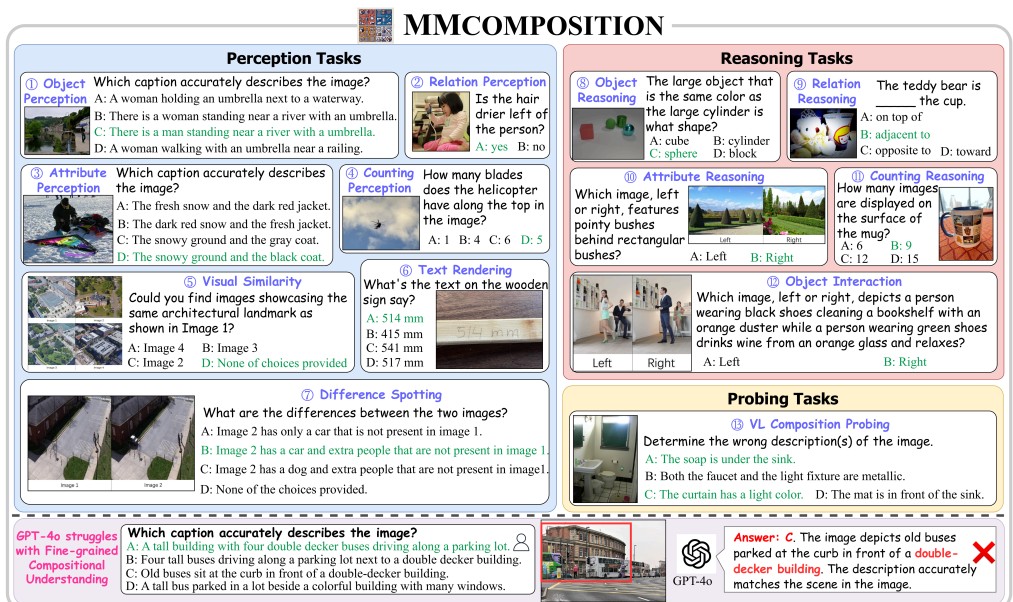

Figure 1: MMCOMPOSITION comprises 13 categories of high-quality VL composition QA pairs, covering a wide range of complex compositions. In the example, GPT-4o failed to understand the compositional aspects of the visual and textual components, misidentifying a three-story building as a double-decker structure. This misinterpretation highlights the limitations of current VLMs.

reasoning abilities. Additionally, some existing benchmarks (Yuksekgonul et al., 2022; Hsieh et al., 2024; Zhao et al., 2022; Thrush et al., 2022; Ray et al., 2023; Ma et al., 2023) evaluate models' compositionality roughly from the perspective of attribute, relation, and object perception. These benchmarks have limitations in evaluating fine-grained visual composition and reasoning. They mainly focus on image-to-text retrieval tasks, assessing basic object, relation, and attribute recognition but neglecting deeper reasoning about object interactions, counting, and complex compositions. As a result, researchers currently have an incomplete understanding of VLMs' compositionality.

To address these issues, we propose MMCOMPOSITION, a novel, human-annotated, high-quality benchmark for the comprehensive evaluation of VLMs' compositionality. MMCOMPOSITION evaluates the compositionality of VLMs in three main dimensions: VL compositional perception, reasoning, and probing, which are further divided into 13 distinct categories of questions, as illustrated in Figure 1. While previous evaluation benchmarks have primarily focused on text-to-image retrieval, single-choice questions, and open-ended text generation, MMCOMPOSITION introduces a more diverse and challenging set of tasks. The benchmark encompasses 4,342 questions, covering both single-image and multi-image scenarios, as well as single-choice and indefinite-choice formats. This expanded range of tasks is designed to evaluate the complex interplay between vision and language in VLMs more effectively. By incorporating a wider variety of complex composition questions, MMCOMPOSITION provides a more comprehensive and in-depth assessment of models' capabilities in cross-modal compositionality, surpassing the evaluations offered by earlier benchmarks like ARO (Yuksekgonul et al., 2022) and Winoground (Thrush et al., 2022). Table 8 highlights the differences between MMCOMPOSITION and other existing datasets that focus on VL compositionality.

In addition to the new benchmark, we also provide a comprehensive analysis of the models' capabilities in fine-grained VL compositional perception and reasoning. Our experiments show that most SOTA VLMs exhibit deficiencies in compositional understanding. Even GPT-4o, despite its advanced capabilities, struggles with tasks requiring nuanced compositional reasoning. These findings highlight the need for further research and development to enhance the compositional abilities of VLMs. Our benchmark serves as a tool for identifying these gaps and inspiring future improvements in VLM design and training. Moreover, we analyze the critical factors in VLM architecture and training that may influence the compositionality of VLMs. According to the empirical results, we reach three findings: **(1) Visual Encoder Design**: While a mixture-of-encoder architecture can enhance compositionality, adding more encoders does not necessarily improve performance. Moreover, models that encode images with minimal degradation of image quality – preserving the original high resolution

Table 1: Comparison with related VL compositional benchmarks: "Yes/No Ratio" refers to the proportion of yes/no questions, "Fine-grained" indicates whether the data provide detailed breakdowns of VL compositional information, and "IT Mismatch Detec." means "Image Text Mismatch Detection".

| Dataset | Yes/No Ratio | Size | Human Annotation | Multi-Image | Indefinite-Choice | Task | Fine-grained |
|---|---|---|---|---|---|---|---|
| Winoground (Thrush et al., 2022) | - | 400 | ✓ | ✓ | - | Compositional Reasoning | ✗ |
| ARO (Yuksekgonul et al., 2022) | - | 50k | ✗ | ✗ | - | T2I Retrieval | ✗ |
| Sugarcrepe (Hsieh et al., 2024) | - | 7,512 | ✗ | ✗ | - | T2I Retrieval | ✗ |
| VL-Checklist (Zhao et al., 2022) | - | 410k | ✗ | ✗ | - | T2I Retrieval | ✗ |
| Cola (Ray et al., 2023) | - | 1,200 | ✗ | ✗ | - | T2I Retrieval | ✗ |
| FineMatch (Hua et al., 2024a) | - | 49.9k | ✓ | ✗ | - | IT Mismatch Detec. | ✓ |
| GQA (Hudson & Manning, 2019) | 0.774 | 22M | ✗ | ✗ | ✗ | Compositional QA | ✓ |
| **MMCOMPOSITION (ours)** | 0.038 | 4,342 | ✓ | ✓ | ✓ | Compositional QA | ✓ |

and aspect ratio – exhibit superior compositionality compared to those that utilize downsampling during the encoding process. **(2) Language Decoder Size**: Larger language decoders are associated with improved compositionality. **(3) The Volume of Training Data**: Fine-tuning models on more diverse datasets helps mitigate some compositionality limitations, driving more robust compositional understanding. In addition, although GPT-4o includes a powerful language model, we find that **for relatively simple QA tasks, only a small portion of its language capabilities are utilized** (compared to the models outperform GPT-4o, whose language model size is only 70B). **Once the language decoder size reaches a certain threshold (e.g., 34B, 70B), the visual encoder has a more significant impact on the model's compositionality**. We demonstrate in Figure 13 that the downsampling image processing in GPT-4o contributes to its inferior performance. Our experimental analysis highlights the limitations of large-scale VLMs in fine-grained compositional perception and reasoning. Our empirical analysis provides a systematic framework for evaluating and enhancing models' capability, pinpointing areas where large models still struggle.

Our main contributions are three-fold:

- We introduce **MMCOMPOSITION**, a novel, human-annotated, high-quality benchmark designed to evaluate the compositionality of pre-trained VLMs. MMCOMPOSITION assesses compositionality across three dimensions: compositional perception, reasoning, and probing, which are further divided into 13 distinct categories of questions. The benchmark includes a diverse set of 4,342 questions, encompassing both single-image and multi-image scenarios, as well as single-choice and indefinite-choice questions, providing a comprehensive and robust evaluating framework for VLM compositionality.

- We comprehensively evaluate 54 well-known VLMs with MMCOMPOSITION. The empirical results highlight the challenging nature of MMCOMPOSITION, as the highest model accuracy reached only 67.95%, compared to 90.31% for human performance. This evaluation reveals a **substantial gap** between state-of-the-art VLMs and human capabilities and provides insights into the limitations of current VLMs.

- We systematically analyze critical factors in VLM architecture that may influence the compositionality of VLMs, including the size of language decoders, the volume of training data, and the visual encoder design. Furthermore, we provide an interpretable analysis of models' limitations in complex compositional understanding. This analysis identifies critical areas for model improvement and suggests directions for future advancements.

## 2 RELATED WORK

### 2.1 VLM EVALUATION BENCHMARKS

The advent of large-scale VLMs has led to the development of numerous benchmarks designed to evaluate various model capabilities. Among the most commonly evaluated are image captioning (Lin et al.; Onoe et al., 2024; Masry et al., 2022), which tests a VLM's ability to generate natural language descriptions of images; VQA (Antol et al., 2015; Marino et al., 2019; Mathew et al., 2020), which assesses the model's capacity to answer image-based questions by integrating visual perception with language understanding or external knowledge; and Visual Reasoning (Johnson et al., 2017; Suhr et al., 2017), which evaluates a model's understanding of spatial relationships and logical reasoning based on visual input. In recent years, researchers have built benchmarks that aim to evaluate the comprehensive capabilities of VLMs (Li et al., 2023a; Liu et al., 2023b; Yue et al., 2023; Fu et al., 2023; Yu et al., 2024a; Lu et al., 2023; Guan et al., 2024). Although some

benchmarks include QA pairs related to compositional reasoning, such as BLINK (Fu et al., 2024), MMVP (Tong et al., 2024b), and Seed-bench (Li et al., 2023a), these are often mixed with other types of QA pairs, making it challenging to assess a model's compositionality precisely. In contrast, MMCOMPOSITION consolidates and refines existing categories of VL compositionality, offering a diverse set of compositional QA pairs that provide a more precise evaluation of model performance.

## 2.2 COMPOSITIONALITY FOR VISION-LANGUAGE MODELS

Compositional understanding of images and text is a critical capability for VLMs. Research indicates that VLMs struggle to distinguish hard negative examples, i.e., image-text pairs that mismatch in at least one aspect (e.g., attribute, relation, object), as there is little incentive for them to learn compositionality during contrastive pre-training (Yuksekgonul et al., 2022). Hsieh et al. (2024) illustrate that contrastive pre-training with generated hard negative examples can improve models' performance on downstream tasks. Various benchmarks have been proposed to assess the capabilities of VLMs in compositional vision-language perception, including VL-Checklist (Zhao et al., 2022), ARO (Yuksekgonul et al., 2022), FineMatch (Hua et al., 2024a), Sugarcrepe (Hsieh et al., 2024), Crepe (Ma et al., 2023), Cola (Ray et al., 2023), CheckList (Zhao et al., 2022), etc. However, these benchmarks often evaluate models' capabilities from limited perspectives, such as object, attribute, and relation perception, and primarily focus on simple tasks like binary image-to-text retrieval, where models need to select the correct caption from pairs containing a correct and a hard negative caption. Moreover, the aforementioned benchmarks often contain a limited range of relations or attributes (e.g., ARO includes 48 relations and 117 attributes). GQA (Hudson & Manning, 2019) includes a diverse set of QA pairs focused on compositional reasoning, but the majority of the questions (77.74%) are simple Yes/No format. In contrast, MMCOMPOSITION offers a more comprehensive assessment with various compositional scenarios, including multi-image and indefinite choice questions, providing a more comprehensive assessment. Furthermore, MMCOMPOSITION evaluates the robustness in detecting complex relationships, including subtle scene composition, object interactions, and higher-order concepts beyond basic perception.

## 2.3 PRE-TRAINED VISION-LANGUAGE MODELS

Vision-language models (Radford et al., 2021; Liu et al., 2024a; Hua et al., 2024b; Ye et al., 2023; Tang et al., 2024; Chen et al., 2024b; Bi et al., 2024; Li et al., 2022; Tong et al., 2024a) aim to achieve multimodal intelligence by jointly understanding and generating visual and language information. Inspired by the remarkable success of recent large language models (LLMs) (Touvron et al., 2023; Chiang et al., 2023; Hua et al., 2021), researchers are now exploring large VLMs that combine pre-trained visual encoders and language decoders to tackle complex multimodal tasks. Flamingo (Alayrac et al., 2022) and BLIP-2 (Li et al.) are two of the early works that explore the integration of LLMs into vision-language pre-training. These models are trained as VL foundation models. Beginning with LLaVA (Liu et al., 2024a), researchers have used LLM-synthesized instruction-following chat data in VQA format for instruction tuning, achieving significantly improved results (Hua et al., 2024a). Subsequent studies have expanded to explore the broader capabilities of multimodal LLMs (Hu et al., 2023; Guan et al., 2024; Lin et al., 2023; Yu et al., 2024c; Tang et al., 2023). However, these efforts place less emphasis on improving the models' ability to fine-grained compositional perception and reasoning.

## 3 MMCOMPOSITION

### 3.1 DATA CURATION

To ensure a comprehensive and high-quality benchmark, we develop an efficient pipeline for curating VQA data that accurately reflects compositional information.

**Data Collection.** We use various datasets with the potential to construct VL compositional QA pairs as our seed data. This collection includes datasets that contain the description of objects, attributes, relations, and counting, such as VL-CheckList (Zhao et al., 2022), Sugar-Crepe (Hsieh et al., 2024), ARO (Yuksekgonul et al., 2022), Crepe (Ma et al., 2023), and DOCCI (Onoe et al., 2024). Additionally, we incorporate sources that are well-suited for constructing VL compositional

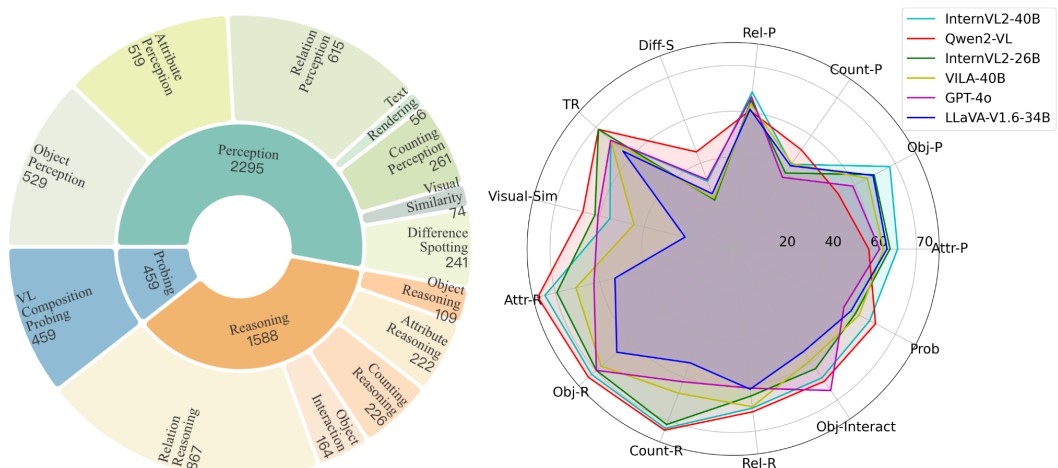

Figure 2: The statistics of 13 distinct categories of QA pairs in MMCOMPOSITION and some models' performance on each category.

reasoning QA pairs, including SVO-Probes (Hendricks & Nematzadeh, 2021), VSR (Liu et al., 2023a), BLINK (Fu et al., 2024), GQA (Hudson & Manning, 2019), Visual Genome (Krishna et al., 2016), and CLVER (Johnson et al., 2017). It also contains datasets with multiple images in each sample, such as Winoground (Thrush et al., 2022), MuriBench (Wang et al., 2024) and NLVR2 (Suhr et al., 2017).

**Question and Answer Construction.** We obtain QA pairs from the seed data in through several methodologies:

For the seed data that only contain positive and negative captions (e.g., ARC (Yuksekgonul et al., 2022)), we first generate sentence embeddings for each caption using Sentence-BERT (Reimers & Gurevych, 2019). We then utilize these embeddings to retrieve the most similar captions from the Visual Genome (Krishna et al., 2016) dataset. This process results in four captions per image in each sample, forming four answer options per question.

For data samples containing multiple images – such as those in the image difference spotting task, which includes two images per question – we concatenate the two images side by side and label them *Left* and *Right* beneath each sub-image. This setup allows for two types of question-answer options: *Left* and *Right* for questions asking which sub-image is described by a caption, and *True* and *False* for questions determining the accuracy of a caption describing the image difference. For tasks that include more than two images per question (e.g., visual similarity assessments), we concatenate all images into a single composite image and label each sub-image as $\text{Image}_1, ..., \text{Image}_i$.

For the probing task, we select several captions from the dense captions in Visual Genome (Krishna et al., 2016) as the correct options and write the misaligned captions manually for the image. Then, we randomly select $x \in \{1, 2, 3, 4\}$ captions from the set of accurate captions for a given image and complement these with $4 - x$ incorrect options drawn from a set of conflict captions. With this approach, we can obtain the indefinite-choice QA pairs.

**Data Filtering and Difficulty Classification.** We divide the data into different difficulty levels: easy, medium, hard, and super hard. To achieve this, we use a voting system with six models, ranging from weaker to stronger, including LLaVA-1.5-13B (Liu et al., 2024b), LLaVA-1.6-Mistral-7B (Liu et al., 2024a), LLaVA-1.6-Vicuna-13B (Liu et al., 2024a), Phi-3-Vision-128K-Instruct (Abdin et al., 2024), InternVL-Chat-V1.5 (Chen et al., 2024b), and Qwen-VL-Chat (Bai et al., 2023). Based on the accuracy of model predictions for each question, questions are categorized into different difficulty levels. Questions with zero correct predictions are classified as super hard, those with one or two correct predictions are labeled as hard, questions with three or four correct predictions are considered medium, and those with more than five correct predictions are categorized as easy. The overall difficulty of the dataset is then controlled by adjusting the ratio of questions at each difficulty level.

**Human Annotation.** All QA pairs in the benchmark are human-annotated. Annotators first assess image quality to ensure it meets the required standards. For human-created data, annotators are first trained with detailed instructions to develop a thorough understanding of the compositional aspects in our dataset. During annotation, they generate QA pairs based on the provided aspect prompts. For GPT-synthesized data sourced from DOCCI, annotators verify whether the question accurately reflects the compositional information in the image and whether the answer appropriately corresponds to the question.

## 3.2 Evaluation Metric

Let $\mathcal{D} = \{\mathcal{D}_m = \{\mathcal{T}_t\}_{t=1}^{T_d}\}_{m=1}^{|D|}$ denotes our dataset , where each catagory $\mathcal{D}_m$ consists of $\mathcal{T}_d$ subtasks. For each subtask, we calculate the accuracy across all annotations. For each question $q \in \mathcal{D}$, let $\mathcal{A}_q$ be the set of correct options, $\mathcal{P}_q$ be the set of predicted (selected) options. The score for question $q$, denoted as $s_q$, is calculated as:

$$s_q = \begin{cases} 1, & \text{if } \mathcal{P}_q = \mathcal{A}_q \\ \frac{|\mathcal{P}_q|}{|\mathcal{A}_q|}, & \text{if } \mathcal{P}_q \subset \mathcal{A}_q \\ 0, & \text{otherwise} \end{cases}$$

Here, $|\cdot|$ denotes the number of options selected by the participant and the number of correct options, $\mathcal{P}_q \subset \mathcal{A}_q$ means all selected options are correct, but some correct options are missing (under-selection). The "otherwise" case covers instances where incorrect options are selected (wrong or over-selection). This equation applies to both the single-choice and indefinite-choice questions. The final weighted average accuracy across all categories is calculated as $\text{ACC} = \sum_{m=1}^{|D|} \sum_{t=1}^{T_d} s_q \times |\mathcal{T}_t|/|\mathcal{D}_d|$, where $|\cdot|$ is the question number in one set.

## 3.3 Quantitative Analysis

MMCOMPOSITION contains 13 different VL composition tasks, including Attribute Perception (**Attr-P**), Object Perception (**Obj-P**), Counting Perception (**Count-P**), Relation Perception (**Rel-P**), Difference Spotting (**Diff-S**), Text Rendering (**TR**), Visual Similarity (**Visual-Sim**), Attribute Reasoning (**Attr-R**), Object Reasoning (**Obj-R**), Counting Reasoning (**Count-R**), Relation Reasoning (**Rel-R**), Object Interaction (**Obj-Interact**), and Compositional Probing (**Prob**). We use GPT-4o to label each question category via in-context learning, followed by manual verification for accuracy. Figure 4 illustrates the difficulty distribution of MMCOMPOSITION, highlighting the challenging nature of our dataset. Figure 5 depicts the distribution of option counts per question, with over half of the data containing more than four options. To analyze the impact of input resolution on model performance, we further display the resolution distribution of images in Figure 6, which reflects the image quality of our data. For textual analysis, we visualize the phrase distribution of questions using a word cloud diagram in Figure 7, clearly depicting the word frequency and distribution across the questions. We also provide a detailed explanation for these 13 categories in Section A.3.

## 4 Revisiting the Compositionality of Pre-trained Vision-Language Models

In this section, we quantify and explore the compositionality of state-of-the-art VLMs and provide a comprehensive evaluation of VLMs. For all experiments, we use a consistent prompt template and the official default hyperparameters for each model.

**Overall performance.** The overall performance indicates that models struggle with perceiving and reasoning about fine-grained VL compositional information. The best human expert achieves an accuracy of 90.31%, significantly outperforming all the models reported in the table. This demonstrates the still existing gap between human expertise and the performance of current models on the MMCOMPOSITION benchmark. This reflects the benchmark's rigorous standards. The open-source InternVL2 (Chen et al., 2024c) series models secured first and second place on the leaderboard. InternVL2-40B performs better than InternVL2-76B. Among the API-based models, Qwen2-VL and GPT-4o achieved the best and second best performance. The superior performance of open-source models with relatively smaller language models compared to GPT-4o, which has a larger language model, is due to their more effective visual encoders. The mean accuracy of 7B and 13B open-source

Table 2: The comprehensive performance of 54 VLMs on Acc, including open source models and API-based models . The **best** and second best results are in bold and underlined, respectively.

| Method | Perception↑ | | | | | | | Reasoning↑ | | | | | Probing↑ | Overall↑ |
|---|---|---|---|---|---|---|---|---|---|---|---|---|---|---|
| | Attr-P | Obj-P | Count-P | Rel-P | Diff-S | TR | Visual-Sim | Attr-R | Obj-R | Count-R | Rel-R | Obj-Interact | Prob | |
| Human | 97.94 | 98.04 | 93.06 | 92.00 | 79.02 | 85.71 | 86.54 | 91.20 | 78.83 | 100.00 | 77.35 | 88.00 | 91.84 | 90.31 |
| InternVL2-40B (Chen et al., 2024b) | 72.22 | 77.69 | 45.21 | 72.53 | 31.12 | 73.21 | 48.65 | 83.78 | 82.57 | 84.51 | 69.20 | 65.85 | 59.59 | 67.95 |
| InternVL2-76B (Chen et al., 2024b) | 70.65 | 76.75 | 48.28 | 70.00 | 19.09 | 78.57 | 48.65 | 85.14 | 83.49 | 85.40 | 70.93 | 67.07 | 58.46 | 67.28 |
| Qwen2-VL-72B (team, 2024) | 59.57 | 51.80 | 52.49 | 62.52 | 45.23 | 82.14 | 67.57 | 87.84 | 84.40 | 84.51 | 71.51 | 70.12 | 69.57 | 65.24 |
| InternVL-Chat-V1.2-Plus (Chen et al., 2024b) | 69.81 | 66.73 | 43.68 | 69.02 | 31.12 | 78.57 | 28.38 | 78.83 | 77.98 | 80.53 | 67.13 | 60.98 | 65.80 | 64.94 |
| InternVL2-26B (Chen et al., 2024b) | 68.46 | 69.57 | 40.23 | 66.96 | 22.82 | 80.36 | 62.16 | 79.28 | 79.82 | 81.86 | 64.55 | 63.41 | 52.43 | 63.08 |
| VILA-40B (Lin et al., 2024) | 65.70 | 66.16 | 45.21 | 63.65 | 23.65 | 75.00 | 44.59 | 70.72 | 77.06 | 67.26 | 69.32 | 59.15 | 62.16 | 62.38 |
| GPT-4o (Achiam et al., 2023) | 63.97 | 58.98 | 37.93 | 66.76 | 32.37 | 82.14 | 60.81 | 62.61 | 79.82 | 61.95 | 61.13 | 75.00 | 54.65 | 59.71 |
| InternVL-Chat-V1.2 (Chen et al., 2024b) | 64.58 | 64.08 | 41.38 | 62.98 | 25.73 | 76.79 | 29.73 | 63.06 | 71.56 | 61.06 | 63.44 | 65.24 | 60.71 | 59.61 |
| InternVL-Chat-V1.5 (Chen et al., 2024b) | 59.44 | 62.38 | 38.31 | 60.47 | 21.58 | 76.79 | 51.35 | 77.93 | 83.49 | 78.32 | 62.05 | 63.41 | 57.01 | 59.58 |
| InternVL2-8B (Chen et al., 2024b) | 62.68 | 61.44 | 31.80 | 59.54 | 25.31 | 73.21 | 33.78 | 78.83 | 75.23 | 73.89 | 62.05 | 62.20 | 54.10 | 58.47 |
| LLaVA-V1.6-34B (Liu et al., 2024a) | 67.24 | 69.00 | 44.06 | 61.31 | 25.73 | 76.79 | 21.62 | 63.15 | 67.89 | 53.10 | 61.59 | 54.27 | 58.17 | 58.25 |
| MiniCPM-V2.6 (Yao et al., 2024) | 65.19 | 61.06 | 41.00 | 61.80 | 21.99 | 73.21 | 37.84 | 63.96 | 73.39 | 68.14 | 55.25 | 60.98 | 54.43 | 57.01 |
| InternLM-XComposer2-4KHD-7B (Dong et al., 2024b) | 62.24 | 58.03 | 39.08 | 58.36 | 23.65 | 67.86 | 27.03 | 70.72 | 74.31 | 60.18 | 58.71 | 59.15 | 60.02 | 56.69 |
| Qwen-VL-Max (Bai et al., 2023) | 53.76 | 54.82 | 36.40 | 58.67 | 22.82 | 80.36 | 41.89 | 53.60 | 65.14 | 53.98 | 61.36 | 62.80 | 63.87 | 55.18 |
| InternLM-XComposer2.5-7B (Zhang et al., 2024a) | 56.68 | 57.84 | 37.93 | 56.82 | 21.58 | 71.43 | 28.38 | 71.17 | 75.23 | 61.06 | 60.55 | 60.98 | 49.64 | 55.10 |
| Hunyuan-Vision | 61.95 | 65.03 | 37.16 | 58.58 | 26.97 | 76.79 | 36.49 | 61.26 | 72.48 | 56.19 | 54.09 | 59.15 | 45.03 | 54.64 |
| InternLM-XComposer2-VL-7B (Dong et al., 2024a) | 59.18 | 55.39 | 40.23 | 56.91 | 25.31 | 60.07 | 31.08 | 67.57 | 73.39 | 61.06 | 55.02 | 53.66 | 57.15 | 54.62 |
| Gemini-1.5-Pro (Reid et al., 2024) | 55.30 | 53.50 | 39.46 | 57.11 | 24.48 | 67.86 | 55.41 | 59.91 | 74.31 | 50.44 | 56.29 | 65.24 | 49.60 | 53.27 |
| Mini-Gemini-34B (Li et al., 2023b) | 58.35 | 59.17 | 37.93 | 53.70 | 25.31 | 73.21 | 39.19 | 54.50 | 73.39 | 58.41 | 57.90 | 61.59 | 41.79 | 53.06 |
| InternVL2-4B (Chen et al., 2024b) | 53.82 | 55.01 | 31.42 | 52.17 | 18.26 | 73.21 | 25.68 | 77.03 | 71.56 | 72.57 | 56.40 | 55.49 | 41.18 | 52.03 |
| LLaMA-3.2-11B-Vision-Instruct | 54.82 | 58.98 | 36.02 | 55.80 | 30.29 | 69.64 | 29.73 | 50.90 | 67.89 | 51.33 | 53.29 | 60.98 | 49.17 | 52.01 |
| MiniCPM-Llama3-V2.5 (Yao et al., 2024) | 51.93 | 51.23 | 36.40 | 49.88 | 19.92 | 76.79 | 20.27 | 69.37 | 77.06 | 68.14 | 58.13 | 62.20 | 41.79 | 51.54 |
| Mini-Gemini-34B-HD (Li et al., 2023b) | 54.95 | 52.36 | 37.55 | 48.35 | 27.80 | 73.21 | 40.54 | 59.91 | 72.48 | 58.85 | 60.09 | 66.46 | 35.91 | 51.48 |
| Bunny-Llama-3-8B-V (He et al., 2024) | 58.16 | 53.50 | 34.87 | 54.07 | 21.58 | 50.00 | 12.16 | 45.95 | 66.06 | 53.10 | 51.67 | 57.32 | 59.44 | 50.81 |
| Mini-Monkey (Huang et al., 2024) | 52.25 | 59.36 | 26.82 | 52.53 | 26.56 | 73.21 | 18.92 | 68.92 | 65.14 | 59.29 | 52.71 | 50.00 | 42.37 | 50.41 |
| Phi3.5-Vision-Instruct (Abdin et al., 2024) | 55.01 | 48.39 | 30.27 | 52.61 | 25.31 | 67.86 | 31.08 | 45.05 | 63.30 | 53.10 | 56.40 | 53.66 | 54.65 | 50.02 |
| ColgVLM2-Llama3-Chat-19B (Hong et al., 2024) | 57.67 | 54.44 | 34.48 | 51.69 | 38.17 | 57.14 | 48.65 | 50.90 | 65.14 | 47.35 | 44.75 | 59.15 | 50.69 | 49.84 |
| Phi3-Vision-128K-Instruct (Abdin et al., 2024) | 55.30 | 43.86 | 30.27 | 51.31 | 25.31 | 69.64 | 40.54 | 45.05 | 65.14 | 47.79 | 48.79 | 60.37 | 56.75 | 48.52 |
| Yi-VL-34B (AI et al., 2024) | 53.02 | 39.89 | 30.27 | 50.33 | 26.14 | 64.29 | 17.57 | 50.45 | 56.88 | 55.31 | 52.94 | 52.44 | 53.88 | 47.86 |
| Step-1V-32K | 46.11 | 42.16 | 26.44 | 46.25 | 25.31 | 67.86 | 43.24 | 66.67 | 66.97 | 62.83 | 52.13 | 59.76 | 45.46 | 47.64 |
| ConvLLaVA-1024-7B (Ge et al., 2024) | 51.73 | 47.26 | 32.57 | 44.96 | 28.22 | 69.64 | 21.62 | 55.41 | 65.14 | 53.10 | 53.06 | 54.88 | 40.89 | 47.32 |
| Yi-VL-6B (AI et al., 2024) | 51.99 | 45.75 | 30.27 | 49.34 | 25.73 | 60.71 | 20.27 | 45.05 | 51.38 | 52.21 | 51.56 | 51.83 | 48.76 | 46.87 |
| Bunny-3B (He et al., 2024) | 49.97 | 50.66 | 26.82 | 48.79 | 25.73 | 50.00 | 12.16 | 46.40 | 61.47 | 47.79 | 46.14 | 51.22 | 55.08 | 46.32 |
| Bunny-4B-V1.0 (He et al., 2024) | 52.50 | 47.64 | 39.08 | 46.00 | 21.16 | 51.79 | 17.57 | 43.69 | 62.39 | 52.21 | 49.94 | 52.44 | 42.66 | 46.07 |
| LLaVA-HR-13B (Luo et al., 2024) | 50.32 | 42.91 | 35.25 | 39.81 | 32.37 | 66.07 | 27.03 | 45.50 | 60.55 | 45.58 | 51.90 | 57.32 | 48.80 | 46.02 |
| ConvLLaVA-1536-7B (Ge et al., 2024) | 50.03 | 54.82 | 28.35 | 41.25 | 27.39 | 69.64 | 29.73 | 51.35 | 64.22 | 52.21 | 52.13 | 64.02 | 34.20 | 45.52 |
| InternVL2-2B (Chen et al., 2024b) | 43.32 | 54.82 | 26.82 | 45.79 | 22.82 | 67.86 | 17.57 | 63.06 | 58.72 | 49.56 | 46.94 | 53.66 | 38.16 | 45.11 |
| Monkey-Chat (Li et al., 2024) | 49.20 | 49.53 | 24.14 | 47.13 | 16.60 | 69.64 | 13.51 | 51.35 | 58.72 | 44.25 | 46.60 | 51.22 | 48.91 | 44.90 |
| Mini-Gemini-13B (Li et al., 2023b) | 43.71 | 41.21 | 27.20 | 41.63 | 21.58 | 62.50 | 33.78 | 55.86 | 68.81 | 50.44 | 53.06 | 57.32 | 32.28 | 43.74 |
| SliME-7B (Zhang et al., 2024b) | 45.70 | 45.94 | 28.74 | 40.76 | 31.12 | 62.50 | 20.27 | 43.24 | 59.63 | 48.23 | 53.17 | 53.05 | 30.03 | 43.45 |
| INF-LLaVA* (Ma et al., 2024) | 43.19 | 46.69 | 32.95 | 41.92 | 24.48 | 57.14 | 20.27 | 50.00 | 66.06 | 55.31 | 48.33 | 54.27 | 31.41 | 43.32 |
| SliME-8B (Zhang et al., 2024b) | 46.50 | 43.29 | 32.18 | 40.27 | 30.29 | 60.71 | 25.68 | 44.59 | 61.47 | 47.79 | 53.29 | 47.56 | 29.96 | 43.29 |
| INF-LLaVA (Ma et al., 2024) | 45.66 | 42.16 | 27.59 | 47.37 | 33.20 | 57.14 | 32.43 | 46.40 | 60.55 | 42.92 | 44.98 | 54.88 | 35.58 | 43.04 |
| LLaVA-HR-7B (Luo et al., 2024) | 40.46 | 43.67 | 31.42 | 40.43 | 28.22 | 64.29 | 37.84 | 46.85 | 60.55 | 48.67 | 49.25 | 56.71 | 33.04 | 42.73 |
| SliME-13B (Zhang et al., 2024b) | 47.46 | 40.64 | 28.74 | 42.55 | 22.41 | 66.07 | 17.57 | 45.50 | 56.88 | 47.79 | 49.83 | 56.10 | 33.55 | 42.63 |
| ConvLLaVA-768-7B (Ge et al., 2024) | 46.50 | 40.45 | 28.35 | 34.88 | 16.60 | 66.07 | 22.97 | 53.15 | 69.72 | 54.42 | 49.02 | 55.49 | 37.11 | 42.40 |
| InternVL2-1B (Chen et al., 2024b) | 43.13 | 48.02 | 22.99 | 43.29 | 23.64 | 64.29 | 18.92 | 54.05 | 58.72 | 49.12 | 45.91 | 57.93 | 27.89 | 42.06 |
| Mini-Gemini-13B-HD (Li et al., 2023b) | 42.29 | 38.37 | 32.18 | 40.20 | 18.67 | 67.86 | 24.32 | 51.35 | 63.30 | 45.58 | 49.83 | 56.71 | 34.28 | 41.99 |
| Qwen-VL-Chat (Bai et al., 2023) | 41.97 | 36.67 | 25.67 | 39.13 | 16.60 | 67.86 | 16.22 | 41.89 | 61.47 | 53.54 | 47.52 | 58.54 | 41.54 | 41.64 |
| DeepStack-L-HD-Vicuna-7B (Meng et al., 2024) | 43.29 | 37.05 | 28.74 | 35.74 | 18.67 | 60.71 | 17.57 | 46.85 | 60.55 | 45.13 | 46.94 | 59.15 | 35.88 | 40.26 |
| DeepStack-L-Vicuna-7B (Meng et al., 2024) | 45.47 | 41.21 | 27.20 | 36.00 | 21.99 | 60.71 | 18.92 | 42.34 | 56.88 | 42.04 | 46.83 | 50.61 | 30.21 | 39.75 |
| LLaVA-V1.6-Vicuna-13B (Liu et al., 2024a) | 37.09 | 29.30 | 23.75 | 32.43 | 24.90 | 66.07 | 12.16 | 40.99 | 54.13 | 42.04 | 50.29 | 48.78 | 38.16 | 38.03 |
| LLaVA-V1.6-Mistral-7B (Liu et al., 2024a) | 36.55 | 29.68 | 24.14 | 39.24 | 31.12 | 64.29 | 13.51 | 36.94 | 47.71 | 42.04 | 41.18 | 49.39 | 38.24 | 37.18 |
| LLaVA-V1.5-13B (Liu et al., 2024a) | 30.92 | 28.36 | 28.35 | 29.89 | 29.46 | 64.29 | 14.86 | 45.50 | 46.79 | 37.17 | 43.60 | 46.34 | 41.39 | 36.07 |
| Random Choice | 23.12 | 23.63 | 21.84 | 25.85 | 29.46 | 35.71 | 25.68 | 36.94 | 46.79 | 38.50 | 35.64 | 47.65 | 28.61 | 30.15 |

VLMs hovers around 36–38%. For reference, we provide the random guess accuracy (30.15%) as a lower bound for the benchmark.

**The tasks where VLMs exhibit relative strengths and weaknesses.** From Table 2, we observe that VLMs perform relatively better on tasks such as Attribute, Object, and Relation Perception, as well as Attribute, Object, and Count Reasoning, where they perform much better than other categories. However, they struggle with tasks such as Count Perception, Difference Spotting, Visual Similarity, and Probing (see illustrations in Fig. 1). These tasks often involve multiple images, some with extreme aspect ratios, and the probing tasks include indefinite-choice questions, which pose additional challenges for the models. GPT-4o performs relatively weaker on Obj-P, Count-P, Attr-R, Count-R, and Rel-R tasks compared to smaller models that outperform it, aligning with the limitations outlined in the official GPT-4o documentation. Overall, the models perform relatively well on mid-level perception and reasoning tasks.

# 5 DIAGNOSTIC ANALYSIS OF FACTORS INFLUENCING MODEL COMPOSITIONALITY

In this section, we analyze the factors that may influence the compositionality of VLMs. We focus on three dominant factors: visual encoder design, language decoder size, and training data volume.

## 5.1 VISUAL ENCODER DESIGN

**High-resolution visual encoders.** A common approach to enhance a model's capability to perceive fine-grained visual content is to introduce higher-resolution encoders. In this study, we employ the control variable method, where the input resolution of the encoders is the only variable, while the training data and text decoders remain fixed. From Table 3, we observe that models with higher resolution encoders demonstrate superior capability for multimodal compositional perception and

reasoning. However, for the Mini-Gemini series models, the introduction of a high-resolution encoder with a patch info mining mechanism unexpectedly resulted in a performance decline.

Table 3: Performance comparison of models with and without high-resolution encoders (Avg. refers to average resolution).

| Method | Resolution | Visual Tokens | Perception Avg. 1055*813 | Reasoning Avg. 935*535 | Probing Avg. 849*530 | Overall |
|---|---|---|---|---|---|---|
| ConvLLaVA-768-7B | 768 | 144 | 36.51 | 52.46 | 37.11 | 42.40 |
| ConvLLaVA-1024-7B (Ge et al., 2024) | 1024 | 256 | $43.70_{+7.19}$ | $54.41_{+1.95}$ | $40.89_{+3.78}$ | $47.32_{+4.92}$ |
| ConvLLaVA-1536-7B | 1536 | 576 | $41.84_{+5.33}$ | $54.09_{+1.63}$ | $34.20_{-6.69}$ | $45.52_{+3.12}$ |
| LLaVA-1.5-13B (Liu et al., 2024a) | 336 | 576 | 29.91 | 43.45 | 41.39 | 36.07 |
| LLaVA-HR-13B (Luo et al., 2024) | 1024 | 1024 | $41.83_{+11.92}$ | $51.26_{+7.81}$ | $48.80_{+7.41}$ | $46.02_{+9.95}$ |
| DeepStack-L-Vicuna-7B (Meng et al., 2024) | 672 | 2880 | 36.92 | 46.60 | 30.21 | 39.75 |
| DeepStack-L-HD-Vicuna-7B (Meng et al., 2024) | 1344 | 14400 | $35.19_{-1.73}$ | $48.87_{+2.27}$ | $35.88_{+5.67}$ | $40.26_{+0.51}$ |
| Mini-Gemini-13B (Li et al., 2023b) | 768 | 576 | 38.51 | 54.60 | 32.28 | 43.74 |
| Mini-Gemini-13B-HD (Li et al., 2023b) | 1536 | 576 | $37.24_{-1.27}$ | $51.07_{-3.53}$ | $34.28_{+2.00}$ | $41.99_{-1.75}$ |
| Mini-Gemini-34B (Li et al., 2023b) | 768 | 576 | 51.25 | 58.94 | 41.79 | 53.06 |
| Mini-Gemini-34B-HD (Li et al., 2023b) | 1536 | 576 | $47.73_{-3.52}$ | $61.40_{+2.46}$ | $35.91_{-5.88}$ | $51.48_{-0.58}$ |

**Mixture-of-encoder.** Another approach to enhancing visual encoders is the use of a mixture-of-encoder architecture. In this setup, image features are extracted by a combination of high-resolution and low-resolution encoders, providing rich visual information to the language decoders. We analyze the relationship between the mixture-of-encoder architecture and model performance by aggregating different encoders while keeping the training data and decoders fixed. We use the LLaVA-1.5 pretraining data for stage-1 pretraining and the EAGLE 1.8M dataset (Bi et al., 2024) for stage-2 fine-tuning. The initial encoder is a CLIP model with 448 resolution (Radford et al., 2021), and the decoder is LLaMA-3-8B (Dubey et al., 2024). We scale up the encoders using: (A) ConvNeXt (Liu et al., 2022), (B) SAM (Kirillov et al., 2023), (C) DINOv2 (Oquab et al., 2023), and (D) Pix2Struct (Lee et al., 2023). The empirical results in Table 4 indicate that combining CLIP with encoder A improves the models' performance; however, as the number of visual encoders increases, the models' performance declines.

Table 4: A comparative analysis of various mixture-of-encoder architectures in relation to model compositionality.

| Method | Visual Encoders | Relolution | Perception | Reasoning | Probing | Overall |
|---|---|---|---|---|---|---|
| LLaVA-1.5 (Liu et al., 2024a) | CLIP | 448 | 44.93 | 54.16 | 53.34 | 49.19 |
| LLaVA-1.5+A | CLIP+A | 1024 | $45.90_{+0.97}$ | $53.34_{-0.82}$ | $56.93_{+3.59}$ | $49.82_{+0.63}$ |
| LLaVA-1.5+A+B | CLIP+A+B | 1024 | $45.96_{+1.03}$ | $52.46_{-1.7}$ | $49.02_{-4.32}$ | $48.66_{-0.53}$ |
| LLaVA-1.5+A+B+C | CLIP+A+B+C | 1024 | $43.41_{-1.52}$ | $52.14_{-2.02}$ | $54.21_{+0.87}$ | $47.74_{-1.45}$ |
| LLaVA-1.5+A+C+D | CLIP+A+C+D | 1024 | $44.90_{-0.03}$ | $51.57_{-2.59}$ | $54.86_{+1.55}$ | $48.39_{-0.90}$ |

**Visual encoder has a more significant impact on the model's compositionality, while GPT-4o struggles with processing higher-resolution images**. By summarizing the empirical results of this study, we find that for relatively simple QA tasks, only a small portion of its language capabilities are utilized (compared to the models outperforming GPT-4o, whose language model size is only 70B). Once the language decoder size reaches a certain threshold (e.g., 34B, 70B), the visual encoder plays a more critical role in the models' performance. As discussed in Section A.2, Qwen2VL processes images by largely preserving their original resolution and aspect ratio. The Internvl-2 series models employ a dynamic 'any-resolution' encoding strategy: images are first mapped to an optimal aspect ratio from predefined ratios, then divided into $448 \times 448$ pixel tiles, with each tile converted into 256 image tokens. These approaches enable the encoders to handle images of any resolution and aspect ratio with minimal degradation of image quality. In contrast, GPT-4o processes images with downsampling when the image's longest side > 2048px or shortest side > 768px (our data contains 889 such examples), contributing to its inferior performance compared to other open-source models.

## 5.2 THE VOLUME OF TRAINING DATA

The volume of training data is a crucial factor influencing models' performance. In this study, we conduct a comparison analysis of this factor. In Table 5, we observe a significant performance increase when the training data is scaled up substantially. For instance, InternVL-Chat-V1.2 and InternVL-Chat-V1.2-Plus, which use 10 times more training data than the former, show significant performance improvements.

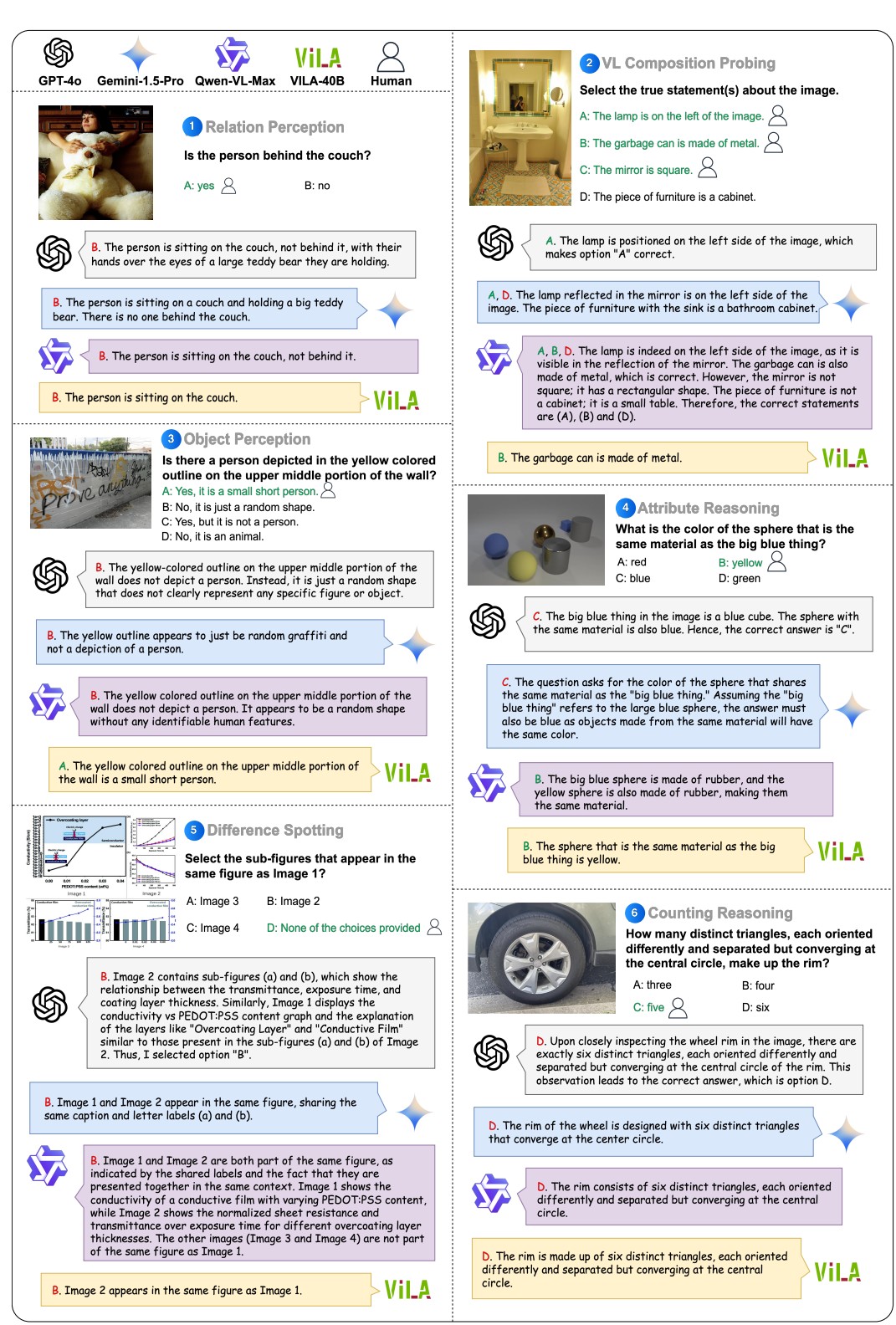

Figure 3: Interpretable analysis of different VLMs. Green letters indicate correct answers, while red letters represent wrong (predicted) answers.

Table 5: The comparison of models with and without training data scale up.

| Method | Dataset Size | Perception | Reasoning | Probing | Overall |
|---|---|---|---|---|---|
| INF-LLaVA (Ma et al., 2024) | 1.25M | 41.80 | 46.98 | 35.58 | 43.04 |
| INF-LLaVA* (Ma et al., 2024) | 2.56M | $40.13_{-1.67}$ | $51.39_{+4.41}$ | $31.41_{-4.17}$ | $43.32_{+0.28}$ |
| InternVL-Chat-V1.2 (Chen et al., 2024c) | 1.2M | 56.49 | 63.79 | 60.71 | 59.61 |
| InternVL-Chat-V1.2-Plus (Chen et al., 2024c) | 12M | $60.73_{+4.24}$ | $70.78_{+6.99}$ | $65.80_{+5.09}$ | $64.94_{+5.33}$ |
| InternVL-Chat-V1.5 (Chen et al., 2024b) | – | 54.14 | 68.20 | 57.01 | 59.58 |
| InternVL2-26B (Chen et al., 2024b) | – | $60.40_{+6.26}$ | $70.03_{+0.83}$ | $52.43_{-4.58}$ | $63.08_{+3.5}$ |

## 5.3 LANGUAGE DECODER SIZE

From Table 2, we observe that models with larger decoders demonstrate stronger performance. To analyze this relationship more accurately, we compare models with different decoder sizes while keeping the encoder and training data constant. The results are shown in Table 6, from which we can conclude that larger language decoders result in better performance.

Table 6: The comparison analysis of text decoder size and models' compositionality.

| Method | Decoder | Perception | Reasoning | Probing | Overall |
|---|---|---|---|---|---|
| InternVL2-1B (Chen et al., 2024b) | Qwen2-0.5B-Instruct | 39.65 | 49.62 | 27.89 | 42.06 |
| InternVL2-2B (Chen et al., 2024b) | InternLM2-Chat-1.8B | $42.37_{+2.72}$ | $51.07_{+1.45}$ | $38.10_{+10.21}$ | $45.11_{+3.05}$ |
| InternVL2-4B (Chen et al., 2024b) | Phi3-Mini-128K-Instruct | $46.94_{+7.31}$ | $62.53_{+12.91}$ | $41.18_{+13.29}$ | $52.03_{+9.97}$ |
| InternVL2-8B (Chen et al., 2024b) | InternLM2.5-Chat-7B | $53.44_{+13.79}$ | $67.00_{+17.38}$ | $54.10_{+26.21}$ | $58.47_{+16.41}$ |
| InternVL2-26B (Chen et al., 2024b) | InternLM2-Chat-20B | 60.40 | 70.03 | 52.43 | 63.08 |
| InternVL2-40B (Chen et al., 2024b) | Nous-Hermes-2-Yi-34B | $65.44_{+5.04}$ | $73.99_{+3.96}$ | $59.59_{+7.16}$ | $67.95_{+4.87}$ |
| InternVL2-76B (Chen et al., 2024b) | Hermes-2-Theta-Llama-3-70B | $63.41_{+3.01}$ | $75.44_{+5.41}$ | $58.46_{+6.03}$ | $67.28_{+4.20}$ |
| LLaVA-V1.6-Mistral-7B (Liu et al., 2024a) | Mistral-7B-Instruct | 33.64 | 42.00 | 38.24 | 37.18 |
| LLaVA-V1.6-Vicuna-13B (Liu et al., 2024a) | Vicuna-13B-V1.5 | $31.15_{-2.49}$ | $47.92_{+5.92}$ | $38.16_{-0.08}$ | $38.03_{+0.85}$ |
| LLaVA-V1.6-34B (Liu et al., 2024a) | Nous-Hermes-2-Yi-34B | $57.82_{+24.18}$ | $58.88_{+16.88}$ | $58.17_{+19.93}$ | $58.25_{+21.07}$ |
| Mini-Gemini-13B (Li et al., 2023b) | Vicuna-13B-V1.5 | 38.51 | 54.60 | 32.28 | 43.74 |
| Mini-Gemini-34B (Li et al., 2023b) | Nous-Hermes-2-Yi-34B | $51.25_{+12.74}$ | $58.94_{+4.34}$ | $41.79_{+9.51}$ | $53.06_{+9.32}$ |
| SliME-7B (Zhang et al., 2024b) | Vicuna-7B-V1.5 | 40.56 | 51.51 | 30.03 | 43.45 |
| SliME-8B (Zhang et al., 2024b) | Llama-3-8B-Instruct | $40.44_{-0.12}$ | $51.26_{-0.25}$ | $29.96_{-3.07}$ | $43.29_{-0.16}$ |
| SliME-13B (Zhang et al., 2024b) | Vicuna-13B-V1.5 | $39.30_{-1.26}$ | $50.06_{-1.45}$ | $33.55_{+3.52}$ | $42.63_{-0.82}$ |
| LLaVA-HR-7B (Luo et al., 2024) | Vicuna-7B-V1.5 | 39.38 | 50.38 | 33.04 | 42.73 |
| LLaVA-HR-13B (Luo et al., 2024) | Vicuna-13B-V1.5 | $41.83_{+2.53}$ | $51.26_{+1.20}$ | $48.80_{+15.25}$ | $46.02_{+3.39}$ |
| Yi-VL-6B (AI et al., 2024) | Yi-6B-Chat | 43.80 | 50.76 | 48.76 | 46.87 |
| Yi-VL-34B (AI et al., 2024) | Yi-34B-Chat | $42.99_{-0.81}$ | $53.15_{+2.39}$ | $53.88_{+5.12}$ | $47.86_{+0.99}$ |

## 5.4 INTERPRETABLE ANALYSIS OF MODEL DEFICIENCIES

We conduct a comprehensive error analysis to better understand the models' deficiencies in fine-grained compositional understanding. In this analysis, the models are required to answer questions and provide explanations in a multi-turn dialogue format. Figures 3, 14, and 15 illustrate the reasons why the models fail to predict the correct answers for each task. For example, in the Obj-P task (example 3), while the 'yellow colored outline' is easily detected by humans, the models struggle to accurately identify the target objects due to the outline being mixed with numerous other characters. Additionally, the models face difficulties with fine-grained object counting, especially when several similar objects are present. In the Count-R (example 6) task, for instance, humans can precisely count the number of triangles on a wheel, but the models confuse the six irregular polygons for triangles.

## 6 CONCLUSION

This paper introduces MMCOMPOSITION, a novel high-quality benchmark for evaluating VLM compositionality. With MMCOMPOSITION, we comprehensively evaluate the compositionality of notable VLMs. Our evaluation reveals a significant gap between these models and human performance, providing insights into the limitations of existing VLMs. Additionally, we systematically analyze factors that may influence compositionality, including visual encoder design, training data volume, and language decoder size. We find that for relatively simple QA tasks, only a small portion of the language model's capacity is utilized (as seen in models outperforming GPT-4o, whose language model has 70B parameters). Once the language decoder reaches a certain size threshold (e.g., 34B, 70B), the visual encoder has a more pronounced impact on compositionality. In summary, our work provides a comprehensive and precise framework for evaluating the compositionality of VLMs, identifies key areas for improvement, and suggests potential directions for future advancements.

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

# A APPENDIX

## A.1 QUANTITIVE RESULTS OF MMCOMPOSITION

In this section, we show statistical results for MMCOMPOSITION in Figure 4 through Figure 7.

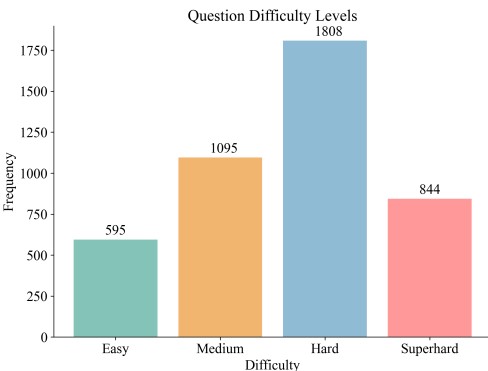

Figure 4: Distribution of difficulty levels across the question set, illustrating the challenging nature of tasks.

Figure 5: Distribution of option counts per question, showing the variety in answer choices provided to evaluate VLMs.

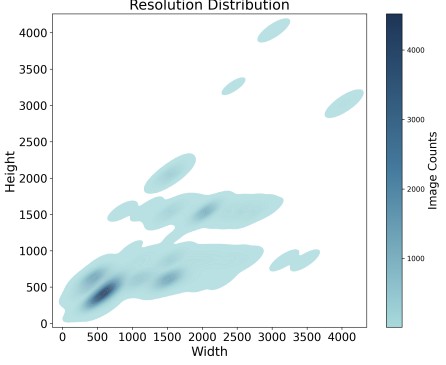

Figure 6: Resolution distribution of images in our benchmark, reflecting the portion of high-quality images in MMCOMPOSITION.

Figure 7: Word cloud of key terms from the questions, illustrating the diversity of compositional content evaluated in the benchmark.

## A.2 COMPARISON ANALYSIS OF IMAGE ENCODING IN GPT-4O, QWEN2-VL, AND INTERNVL-2

In GPT-4o, when the image detail parameters are set to "high", images are first scaled to fit within a $2048 \times 2048$ square while maintaining their aspect ratio. Then, the images are further scaled so that the shortest side is 768px long. Finally, GPT-4o calculates how many 512px squares the image contains, with each square costing 170 tokens. An additional 85 tokens for low resolution are always added to the final total. As a result, GPT-4o does not achieve true "any resolution" image processing.

In Qwen2-VL and InternVL-2, the image encoders adopt a dynamic "any resolution" encoding strategy. The images are first mapped to an optimal aspect ratio from predefined ratios, then divided into $448 \times 448$ or $28 \times 28$ pixel tiles, with each tile converted into 256 or 1 image tokens. A thumbnail is then generated to capture the global context. This allows the encoders to handle images of any resolution and aspect ratio. Furthermore, the image encoder in Qwen2-VL is a 675M ViT with a two-dimensional positional encoding mechanism, while InternVL-2 utilizes the more powerful InternViT with 6B parameters. This distinction contributes to the superior performance of the compositionality

of Qwen2-VL and InternVL-2 in our benchmark. In Table 7, we provide a comparison of the properties of visual encoders for the aforementioned models.

Table 7: Visual encoder comparison of GPT-4o, InternVL2 and Qwen2-VL.

| Method | Visual Encoder | Image Tile Size | Maximum Number of Tiles | Maximum Aspect Ratio | # of Tokens for One Tile |
|---|---|---|---|---|---|
| GPT-4o | - | 512 x 512 | 8 | any | 170 |
| InternVL2 | InternViT-6B | 448 x 448 | 12 | 1:6 | 256 |
| Qwen2-VL | ViT-675M | 28 x 28 | dynamic | any | 1 |

## A.3 DEFINITION OF 13 DISTINCT CATEGORIES IN MMCOMPOSITION

- **Attribute Perception**: The specific attributes or properties of the object perception task that can be solved by humans "within a blink".

- **Object Perception**: Identification or recognition of objects in the image.

- **Counting Perception**: Counting the number of objects or elements in the image.

- **Relation Perception**: Understanding the relationships between objects in the image.

- **Difference Spotting**: Identifying differences or changes between objects or scenes in two similar images.

- **Text Rendering**: Reading or interpreting text present in the image.

- **Visual Similarity**: Comparing similarities between objects or elements across multiple images.

- **Attribute Reasoning**: Identifying and reasoning about specific attributes or properties of objects in the image.

- **Object Reasoning**: Identifying and reasoning about objects in the image.

- **Counting Reasoning**: Identifying and reasoning about the number of objects or elements in the image.

- **Relation Reasoning**: Identifying and reasoning about the spatial arrangement or positioning of objects in the image.

- **Object Interaction**: Understanding interactions among multiple objects in the image.

- **VL Composition Probing**: Examining the composition or combination of visual and textual elements in images, where models are required to accurately find all the complex compositional descriptions about the image.

## A.4 CHARACTERISTICS OF QUESTIONS WHERE MODELS UNDERPERFORM

We define the comprehensive performance value (CPV) for each question as the average score across 54 VLMs. By comparing each question's CPV with the score of a random choice within its class, we find that 1,159 questions have a CPV lower than that of random chance. We show statistical results questions with low CPV in Figure 8 through Figure 11.

## A.5 ANALYSIS OF GPT-4O'S UNDERPERFORMANCE IN SPECIFIC TASKS

Since GPT-4o performs relatively weaker on Obj-P, Count-P, Attr-R, Count-R, and Rel-R tasks compared to smaller models that outperform it, we aim to provide an intuitive analysis of the reasons behind its poor performance on these tasks. Figure 12 presents interpretable examples for the aforementioned categories.

## A.6 MORE INTERPRETABLE EXAMPLES

To provide a clearer and more comprehensive interpretation of the models' capabilities, we present additional interpretable examples in Figure 14 and Figure 15.

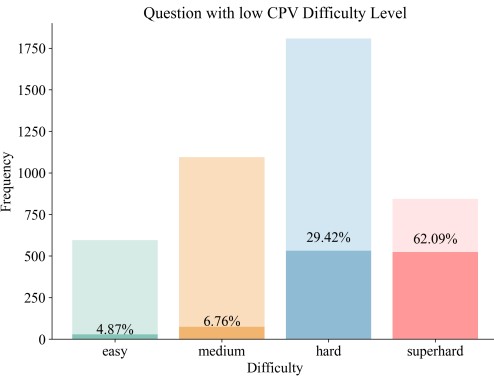

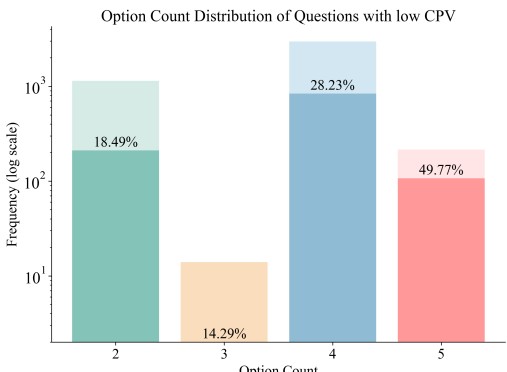

Figure 8: Distribution of difficulty levels of questions with low CPV, illustrating the authenticity of the difficulty distribution in MMCOMPOSITION.

Figure 9: Distribution of option counts for questions with low CPV.

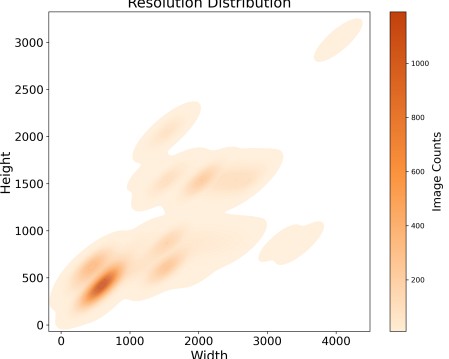

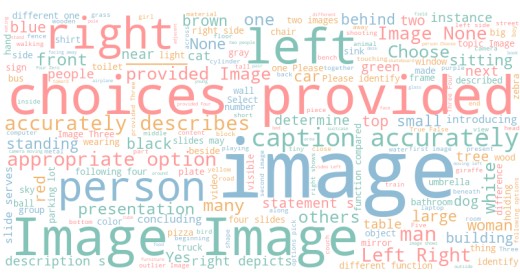

Figure 10: Resolution distribution of images with low CPV.

Figure 11: Word cloud of key terms from the questions with low CPV, illustrating the keywords appearing in the questions that VLMs are hard to answer currently.

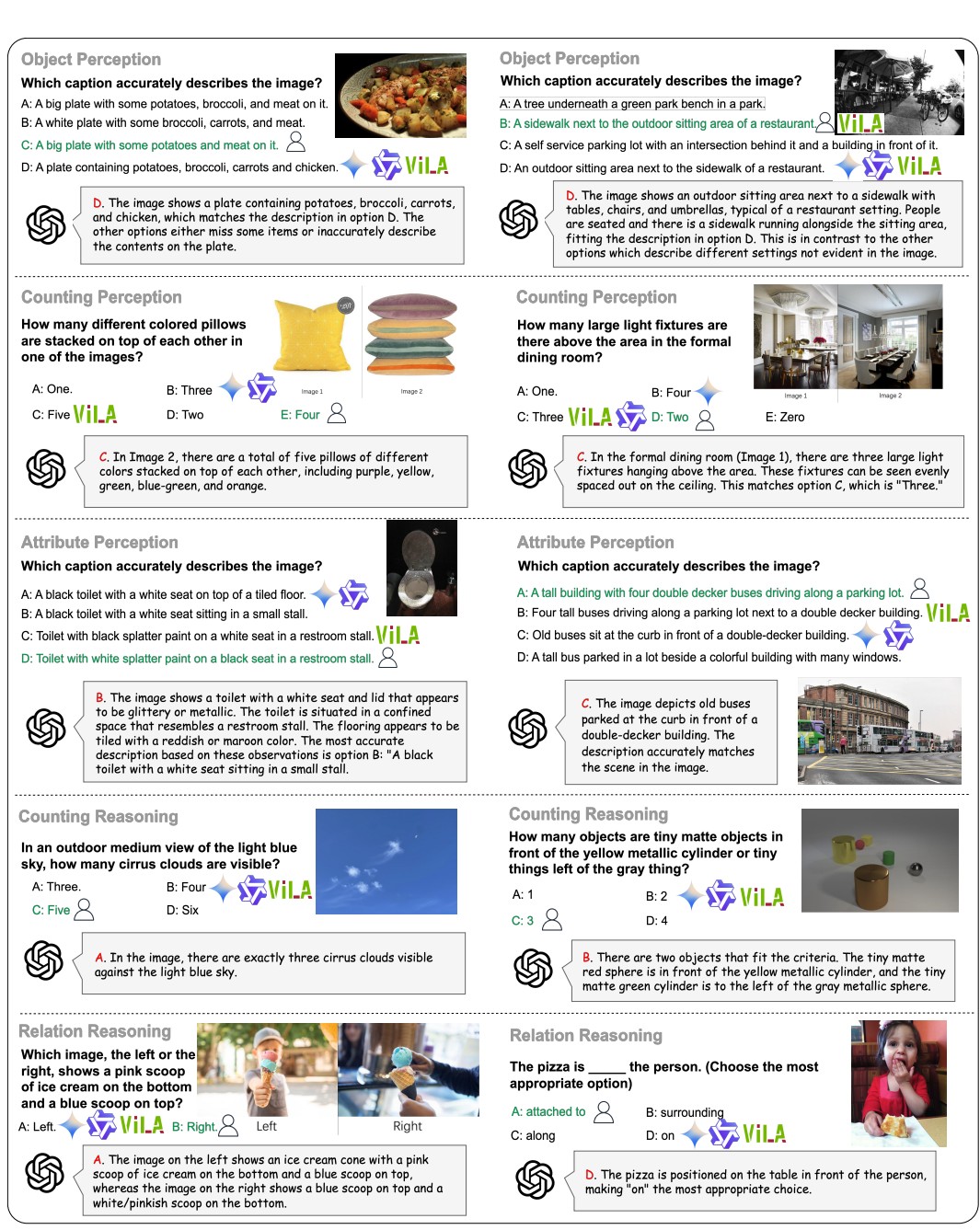

Figure 12: GPT-4o Weak Category Analysis. The logos of the models or human displayed to the right of the option(s) indicate that the model or human has selected the option(s) as the correct answer(s).

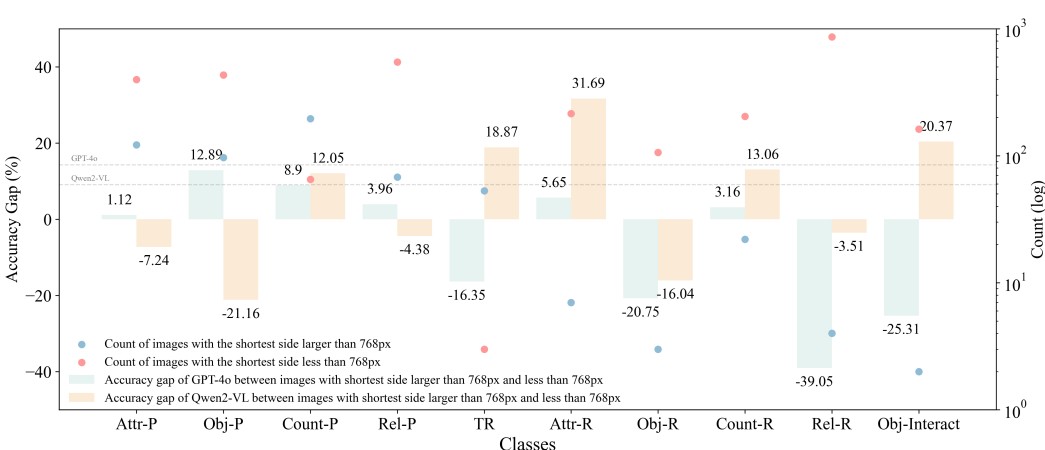

Figure 13: Performance gap between images whose shortest side $> 768px$ and those $\leq 768px$, defined as gap $= Acc_{>768px} - Acc_{\leq 768px}$. The histogram shows the distribution of performance gaps across 13 tasks. The average performance gap for GPT-4o is 14.26, while for Qwen2-VL, it is 9.05. The smaller gap for Qwen2-VL indicates its greater effectiveness in processing high-resolution images. Additionally, Qwen2-VL's performance gaps are more consistently positive across different tasks, further highlighting its robustness in handling high-resolution images.

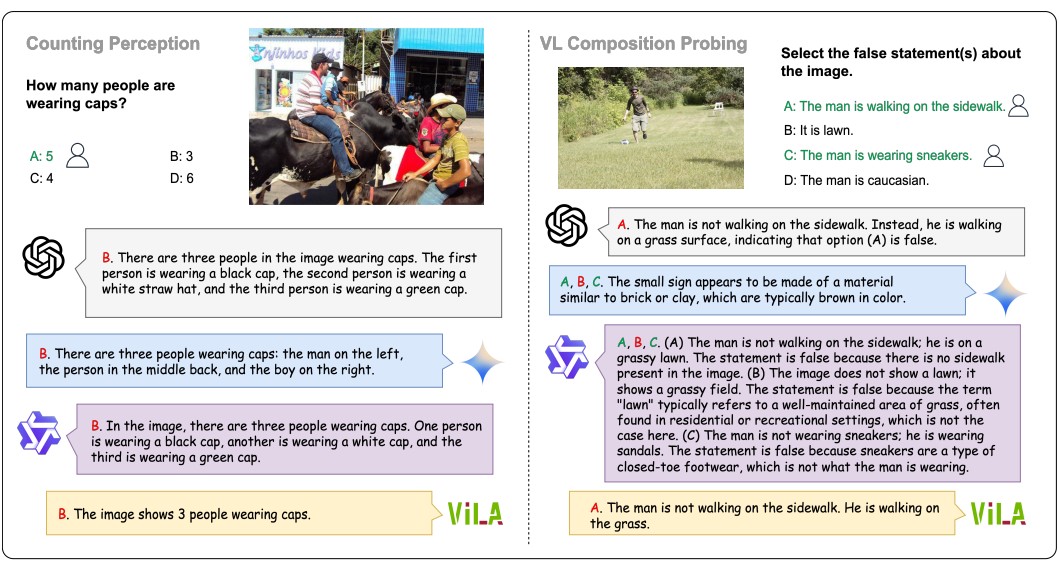

Figure 14: More interpretable analysis of different VLMs. Green indicates correct answers, while red represents the predicted wrong answers.

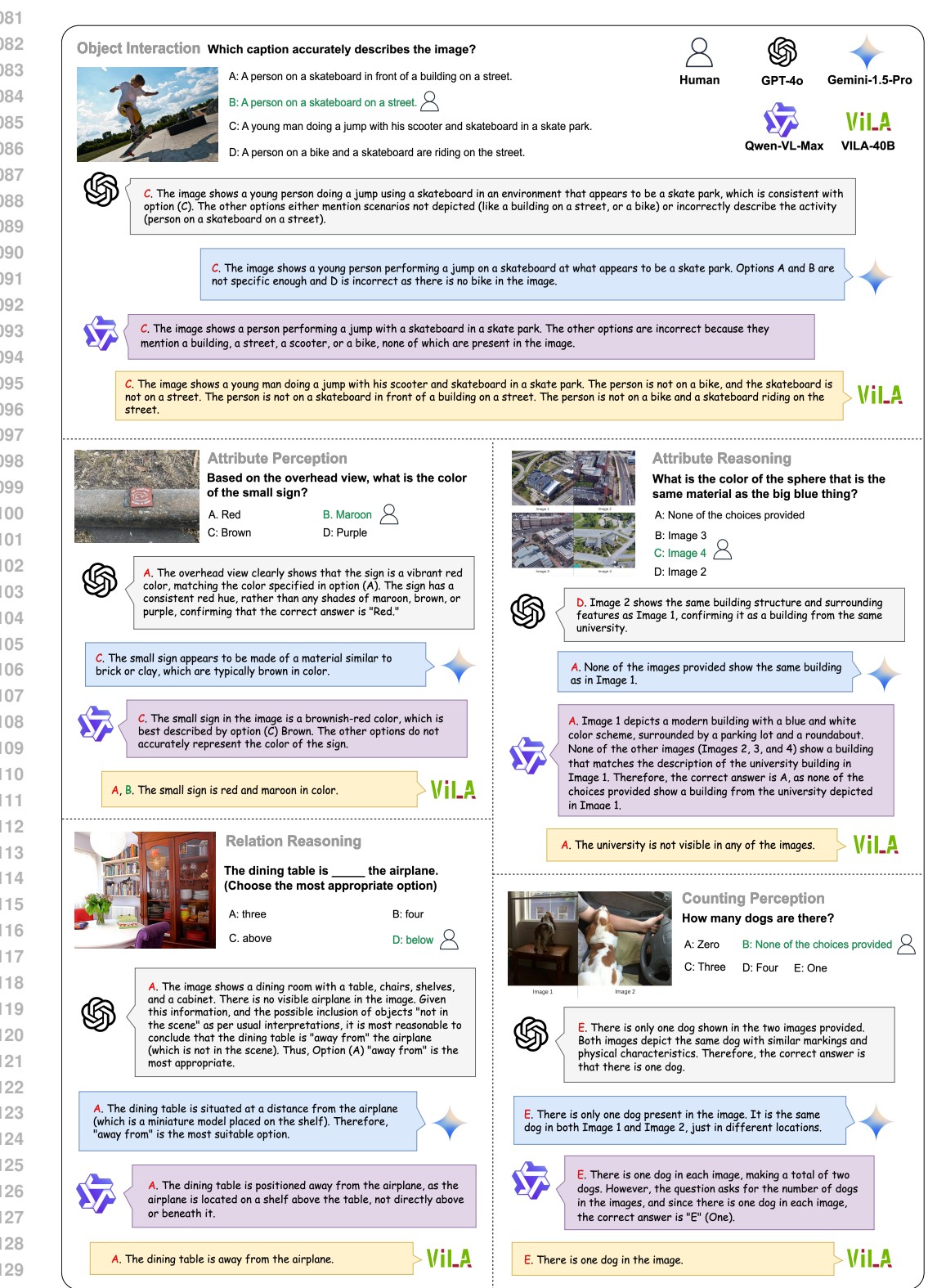

Figure 15: More interpretable analysis of different VLMs. Green indicates correct answers, while red represents the predicted wrong answers.

Table 8: Comparison with related VL benchmarks: "Multi-Hop" refers to whether the dataset contains questions that need multi-hop reasoning, "Comprehensive" in the Capabilities column indicates the benchmark evaluates multiple capabilities for VLMs (e.g., recognition, OCR, knowledge, math, and spatial reasoning).

| Dataset | Size | Human Annotation | Multi-Hop | Capabilities | Best Performance (Model/Human) |
|---|---|---|---|---|---|
| MMBench Liu et al. (2023b) | 3,217 | ✗ | ✗ | Comprehensive | 86.1 / - |
| MME Fu et al. (2023) | 2,800 | ✓ | ✗ | Comprehensive | 1790.04/- |
| MMStar Chen et al. (2024a) | 1,500 | ✓ | ✗ | Comprehensive | 66.0/- |
| SeedBench Liu et al. (2023b) | 19k | ✓ | ✗ | Comprehensive | 72.4 / - |
| MMMU Yue et al. (2023) | 11.5k | ✓ | ✗ | College-Level Subject Knowledge | 69.1 / 88.6 |
| HalBench Guan et al. (2024) | 1,129 | ✓ | ✗ | Hallucination | 67.58 / - |
| **MMCOMPOSITION (ours)** | 4,342 | ✓ | ✓ | Compositionality | 67.95 / 90.31 |

## A.7 ADDITIONAL EXPERIMENTS

To further verify the challenging nature of MMCOMPOSITION and demonstrate the indispensable role of images, we conducted additional experiments under image-blind settings. The results are presented in Table 10. We also conducted experiments to compare different visual-to-language (V2L) adapters and their impact on model performance, as summarized in Table 9. Additionally, we examined the models' abilities to handle multi-hop reasoning questions, the effect of providing in-context examples, and the performance when multiple images are used. These experiments aim to provide a comprehensive understanding of the factors influencing model performance on MMCOMPOSITION.

**Image-blind Setting.** As shown in Table 10, all models experienced significant performance drops across all evaluation dimensions when visual inputs were removed. For instance, the overall score of Qwen2-VL-72B decreased by **20.50%**, underscoring the indispensable role of images in these tasks. This substantial decline confirms that MMCOMPOSITION effectively evaluates the integration of visual and linguistic understanding, as models struggle without visual context.

**V2L Adapters Comparison.** We also compare different V2L adapters. As shown in Table 9, models that utilize an MLP adapter (e.g., LLaVA1.5-13B) generally outperform those with a Q-Former adapter in overall performance. Specifically, LLaVA1.5-13B achieves an overall score of **42.03**, surpassing InstructBLIP-13B's score of **37.06**. This suggests that the choice of adapter architecture significantly influences a model's ability to effectively integrate visual features.

**Multi-hop/Non-multi-hop Question Setting.** We analyze the performance on multi-hop versus non-multi-hop question settings (Table 11) and observe that some models perform better on multi-hop questions, indicating strength in complex reasoning tasks. For example, Qwen2-VL-72B achieved an overall score of **70.22** on multi-hop questions, compared to **58.55** on non multi-hop ones. This demonstrates the model's enhanced capability to handle questions requiring multiple reasoning steps.

**In-context Learning Setting.** The results in Table 12 indicate that in-context examples do not consistently improve performance. For Qwen2-VL-72B, adding one example slightly decreased the overall score by **0.92%**, while adding more examples led to further declines. Similarly, InternVL2-40B experiences a drop of up to **13.44%** with three in-context examples. These results suggest that the models may not effectively utilize in-context learning for visual reasoning tasks, possibly due to limitations in their training data or architectural design.

**Multi-image Setting.** As shown in Table 13, the performance varies between models. Qwen2-VL-72B shows an improvement in overall score by **4.14%** when multiple images are provided, indicating effective utilization of additional visual information. In contrast, InternVL2-40B's performance decreases by **1.62%**, suggesting difficulties in integrating information from multiple images.

These additional experiments reinforce the challenging nature of MMCOMPOSITION and highlight the importance of visual information, adapter architectures, and the complexities involved in multi-hop reasoning and in-context learning within multimodal models. The findings provide valuable insights for future research aiming to enhance the capabilities of vision-language models.

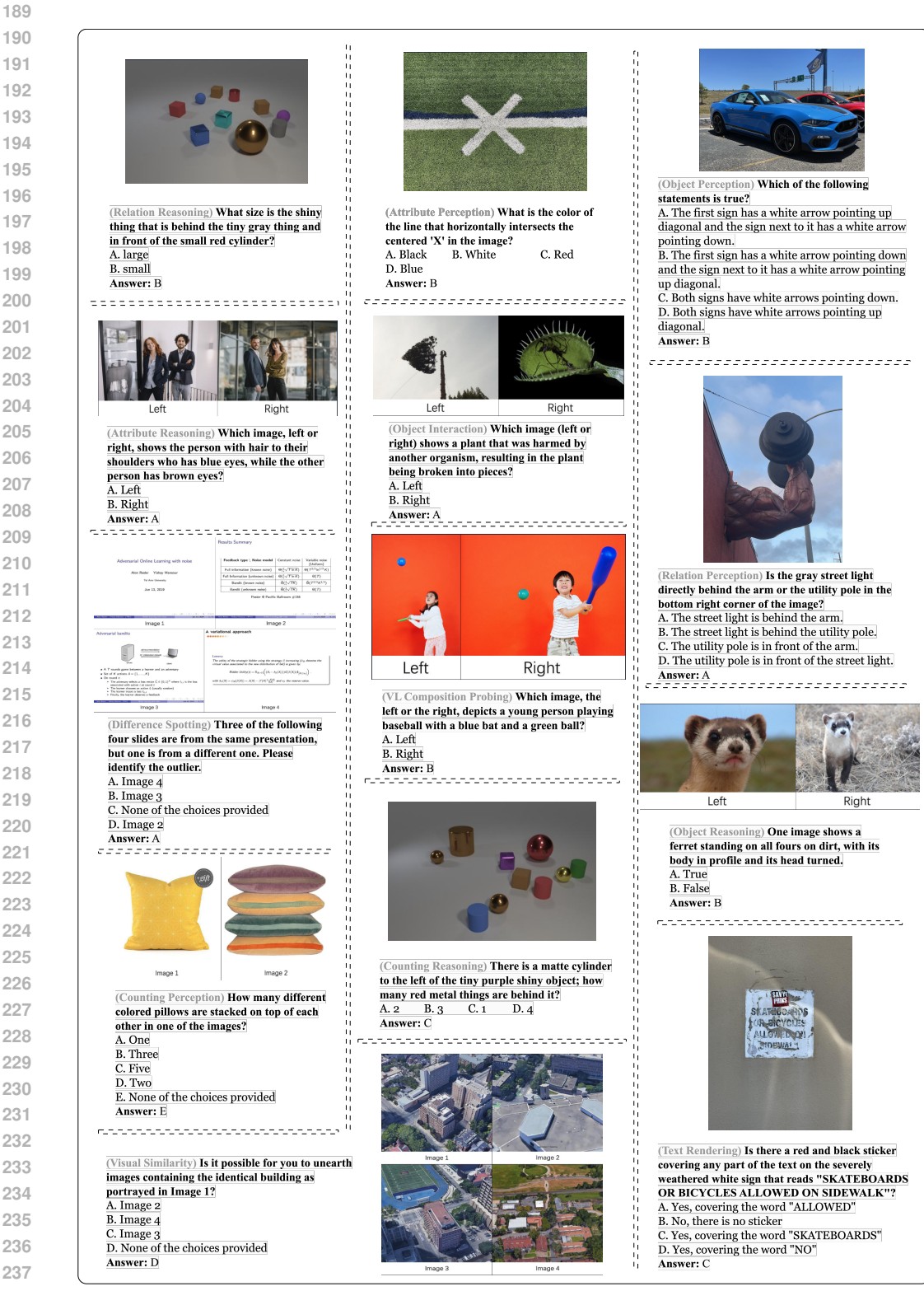

Figure 16: Examples of multi-hop questions: The ratio of multi-hop to non-multi-hop questions in our dataset is 2,459 to 1,841.

Table 9: Comparison of Different Adapters for Model's Performance.

| Model | Visual Encoder | LLM | V2L Adapter | Perception | Reasoning | Probing | Overall |
|---|---|---|---|---|---|---|---|
| mPLUG-Owl2 Ye et al. (2024) | ViT-L/14 | LLaMA2-7B | Q-Former | 36.90 | 46.16 | 30.36 | 39.59 |
| InstructBLIP-7B Dai et al. (2023) | ViT-G/14 | Vicuna-7B | Q-Former | 33.22 | 43.70 | 31.41 | 36.86 |
| LLaVA1.5-7B Liu et al. (2024b) | ViT-L/14 | Vicuna-7B | MLP | 36.51 | 47.04 | 30.32 | 39.71 |
| InstructBLIP-13B Dai et al. (2023) | ViT-G/14 | Vicuna-13B | Q-Former | 35.53 | 42.70 | 25.24 | 37.06 |
| LLaVA1.5-13B Liu et al. (2024b) | ViT-L/14 | Vicuna-13B | MLP | 37.23 | 49.75 | 39.32 | 42.03 |

Table 10: Results for Image-Blind Setting.

| Model | Perception | Reasoning | Probing | Overall |
|---|---|---|---|---|
| Qwen2-VL-72B | 56.53 | 76.39 | 70.26 | 65.24 |
| Qwen2-VL-72B-blind | $45.16_{-11.37}$ | $48.17_{-28.22}$ | $30.76_{-39.50}$ | $44.74_{-20.50}$ |
| InternVL2-26B | 60.40 | 70.03 | 52.43 | 63.08 |
| InternVL2-26B-blind | $34.80_{-25.60}$ | $42.63_{-27.40}$ | $32.17_{-20.26}$ | $37.39_{-25.69}$ |
| InternVL2-40B | 64.57 | 74.12 | 67.14 | 67.95 |
| InternVL2-40B-blind | $37.88_{-26.69}$ | $43.35_{-30.77}$ | $34.28_{-32.86}$ | $39.54_{-28.41}$ |
| InternVL2-76B | 63.41 | 75.44 | 58.46 | 67.28 |
| InternVL2-76B-blind | $33.93_{-29.48}$ | $44.08_{-31.36}$ | $32.68_{-25.78}$ | $37.51_{-29.77}$ |

Table 11: Comparison of models' performance on multi-hop and non multi-hop questions.

| Model | Perception | Reasoning | Probing | Overall |
|---|---|---|---|---|
| InternVL2-40B-non-multi-hop | 74.11 | 66.52 | - | 72.28 |
| InternVL2-40b-multi-hop | 51.24 | 77.01 | 59.59 | 64.63 |
| Qwen2-VL-72B-non-multi-hop | 55.05 | 69.37 | - | 58.55 |
| Qwen2-VL-72B-multi-hop | 58.91 | 79.22 | 69.57 | 70.22 |
| VILA-40B-non-multi-hop | 66.29 | 61.49 | - | 65.14 |
| VILA-40B-multi-hop | 44.58 | 71.62 | 62.16 | 60.25 |
| GPT-4o-non-multi-hop | 63.19 | 57.77 | - | 61.90 |
| GPT-4o-multi-hop | 48.51 | 66.76 | 54.65 | 58.03 |
| LLaVA-1.6-34B-non-multi-hop | 66.14 | 61.27 | - | 64.98 |
| LLaVA-1.6-34B-multi-hop | 44.20 | 57.91 | 58.17 | 53.09 |
| Gemini-1.5-Pro-non-multi-hop | 55.68 | 46.61 | - | 53.50 |
| Gemini-1.5-Pro-multi-hop | 42.39 | 62.78 | 49.60 | 53.09 |

Table 12: Results for In-context Setting.

| Model | Perception | Reasoning | Probing | Overall |
|---|---|---|---|---|
| Qwen2-VL-72B | 56.53 | 76.39 | 70.26 | 65.24 |
| Qwen2-VL-72B-1example | $62.19_{+5.66}$ | $73.30_{-3.09}$ | $43.94_{-26.32}$ | $64.32_{-0.92}$ |
| Qwen2-VL-72B-2example | $63.06_{+6.53}$ | $70.84_{-5.55}$ | $46.37_{-23.89}$ | $64.14_{-1.10}$ |
| Qwen2-VL-72B-3example | $61.61_{+5.08}$ | $69.46_{-6.93}$ | $48.87_{-21.39}$ | $63.13_{-2.11}$ |
| InternVL2-40B | 64.57 | 74.12 | 67.14 | 67.95 |
| InternVL2-40B-1example | $54.01_{-10.56}$ | $66.62_{-7.50}$ | $36.97_{-30.17}$ | $56.82_{-11.13}$ |
| InternVL2-40B-2example | $52.37_{-12.20}$ | $65.24_{-8.88}$ | $36.24_{-30.90}$ | $55.37_{-12.58}$ |
| InternVL2-40B-3example | $51.05_{-13.52}$ | $63.73_{-10.39}$ | $39.94_{-27.20}$ | $54.51_{-13.44}$ |

Table 13: Results for Multi-image Setting.

| Model | Perception | Reasoning | Probing | Overall |
|---|---|---|---|---|
| Qwen2-VL-72B | 55.36 | 77.17 | 89.86 | 71.75 |
| Qwen2-VL-72B-multi | $63.01_{+7.65}$ | $80.35_{+3.18}$ | $89.19_{-0.67}$ | $75.89_{+4.14}$ |
| InternVL2-40B | 42.35 | 73.27 | 88.51 | 65.26 |
| InternVL2-40B-multi | $39.29_{-3.06}$ | $72.54_{-0.73}$ | $86.49_{-2.02}$ | $63.64_{-1.62}$ |

