# OpenReview forum: "MMCOMPOSITION: Revisiting the Compositionality of Pre-trained Vision-Language Models"
_ICLR.cc/2025/Conference — ICLR 2025 Conference Withdrawn Submission_

### Official Review · Reviewer_Deoo · 2024-10-27

**Soundness:** 3
**Presentation:** 3
**Contribution:** 3
**Rating:** 6
**Confidence:** 4

**Summary:**

The paper proposes MMComposition - a human-annotated benchmark dataset for evaluation of the compositionality of large Vision-language models.
The benchmark contains 4.3K questions in three main dimensions: perception, reasoning and probing which are divided into 13 categories. There are both questions that contain a single image and multiple images. Most questions have a single correct answer. There are 459 questions with indefinite-choice.
The benchmark demonstrates human performance (90.31%) and state-of-the VLMs (best performance of 67.95% among 54 evaluated VLMs).
There is also analysis of impact of VLM architecture factor on the benchmark performance, e.g. visual encoder design, language decoder size, training data volume.

**Strengths:**

1) The dataset is human-annotated and covers a wide range of tasks in terms of compositional understanding
2) The paper evaluates 54 representative large VLMs including open-source and proprietary ones. The benchmark is challenging and demonstrates large performance gap between human and VLMs.
3) The analysis on model component provides valuable insight on model design.

**Weaknesses:**

1) The paper categories the questions into 4 difficulty levels based on the performance of 6 open-source models. In Figure 8, it shows that 62.09% of questions in the category “superhard” lead to the average performance on all VLMs below the average level. It would be interesting to analyze what characteristics lead to the different difficulty levels of these questions? This can shed light on how to design difficult questions for the competent VLMs.
2) In the evaluation benchmark, for questions that contain multiple images, the images are concatenated into a big collage and fed into the model. Some of the VLMs have multiple-image samples in the training data and can perform VQA with multiple input images. Does it impede the performance of these models to feed the collage into them?

**Questions:**

1) In line 254, “we select several captions from the dense captions in Visual Genome as the correct options and write the misaligned captions manually for the image”
What are the criteria for writing the misaligned captions? In terms of which characteristics do the misaligned captions differ from the original captions?

---

> ### Author Response · Authors · 2024-11-20
>
> Thanks for your constructive suggestions. Your endorsement of our dataset and experiments gives us significant encouragement.
>
> Q1: Thank you for your suggestion! We will analyze the patterns of super hard questions for shed light on how to design difficult questions for the competent VLMs in the revised paper.
>
> Q2: We conducted experiments to compare different formats of input images, including combining multiple images into a single 'super image' and feeding the models with a list of images. The results, shown in the table, demonstrate that these formats have varying impacts on different models' performance. Specifically, providing a list of images improved the performance of Qwen2-VL-72B, but led to a decrease in performance for InternVL2-40B.
>
> Model              | Perception       | Reasoning       | Probing          | Overall         |
> |-|-|-|-|-|
> Qwen2-VL-72B          | 55.36               | 77.17              | 89.86               | 71.75              |
> Qwen2-VL-72B-multi    | 63.01 (+7.65)       | 80.35 (+3.18)      | 89.19 (-0.67)       | 75.89 (+4.14)      |
> InternVL2-40B         | 42.35               | 73.27              | 88.51               | 65.26              |
> InternVL2-40B-multi   | 39.29 (-3.06)       | 72.54 (-0.73)      | 86.49 (-2.02)       | 63.64 (-1.62)      |
>
> Q3: We follow the method proposed in FineMatch [1], which adopts the criteria of replacing attribute, relation, and object phrases while maintaining the Part of Speech (POS) tags unchanged. This approach keeps the mismatched captions as similar as possible at the character level to the original correct captions.
>
> [1] "FineMatch: Aspect-Based Fine-Grained Image and Text Mismatch Detection and Correction." ECCV, 2024.

---

> > ### Comment · Reviewer_Deoo · 2024-11-23
> >
> > Thanks for the efforts on the additional experiments and clarification. My concerns are addressed.

---

### Official Review · Reviewer_SeFC · 2024-10-30

**Soundness:** 3
**Presentation:** 3
**Contribution:** 3
**Rating:** 5
**Confidence:** 3

**Summary:**

The paper "MMCOMPOSITION: Revisiting the Compositionality of Pre-Trained Vision-Language Models" presents MMCOMPOSITION, a new benchmark focused on testing VLMs' ability to handle complex compositional tasks like object interactions, counting, and scene reasoning. With 4,342 annotated questions across 13 categories, the benchmark highlights a clear performance gap between models and humans (67.95% vs. 90.31% accuracy). Results suggest that improving high-resolution encoders, scaling language decoders, and expanding training data are key to better compositional reasoning in VLMs. MMCOMPOSITION offers a practical tool for refining future VLMs to better understand complex compositions.

**Strengths:**

Strengths of MMCOMPOSITION:

1. Targeted Evaluation of Compositionality for VLMs: MMCOMPOSITION provides a focused benchmark to assess compositional reasoning in Vision-Language Models, an area where existing models often fall short. By going beyond basic attribute recognition, MMCOMPOSITION evaluates tasks like multi-image reasoning, object interactions, and counting, all of which are crucial for real-world, nuanced understanding.

2. Improvement upon Existing Compositional Datasets: This benchmark builds on and enhances data from existing compositional datasets, such as ARO, to create a more diverse and challenging evaluation framework. By curating tasks that move beyond traditional benchmarks, MMCOMPOSITION offers a comprehensive dataset for testing complex visual-language interactions.

3. In-Depth Model Comparison and Component Analysis: MMCOMPOSITION evaluates over 50 VLMs across different architectural components, allowing a detailed comparison. This thorough assessment reveals how factors like encoder resolution, decoder size, and training data diversity impact compositional reasoning. It offers practical insights that can guide future improvements in model design.

**Weaknesses:**

typos:
table 4 - Relolution

1. In-context multimodal compositionality: Adding tests for in-context multimodal compositionality could strengthen the benchmark, as this capability is crucial for real-world applications. Evaluating models' ability to maintain compositional understanding across multi-modal inputs, rather than isolated tasks, could enhance the dataset's relevance.
2. Multi-hop compositional problems: The paper would benefit from including multi-hop reasoning tasks, where models must integrate multiple compositional steps to arrive at an answer. This kind of problem is essential for advanced compositionality and would make the benchmark more challenging and comprehensive.
3. Questionable novelty: The novelty of the paper could be improved if it incorporated points 1 and 2. Adding in-context multimodal compositionality and multi-hop compositional problems would make MMCOMPOSITION a more distinctive and valuable benchmark.

**Questions:**

see weaknesses

---

> ### Author Response · Authors · 2024-11-20
>
> Thank you for your time, thorough comments, and valuable suggestions. We are pleased that you acknowledged our dataset as an improvement upon existing compositional datasets and recognized our experiments for their in-depth model comparison and component analysis.
>
>
> Q1: Thank you for your suggestion. We have added the experiments results under the ICL setting. From the table we can observe that while introducing context examples into the prompt, models’ performance decreased in different degrees.
>
> Model                | Perception         | Reasoning          | Probing             | Overall            |
> |-|-|-|-|-|
> Qwen2-VL-72B            | 56.53                 | 76.39                 | 70.26                  | 65.24                 |
> Qwen2-VL-72B-1example   | 62.19 (+5.66)         | 73.30 (-3.09)         | 43.94 (-26.32)         | 64.32 (-0.92)         |
> Qwen2-VL-72B-2example   | 63.06 (+6.53)         | 70.84 (-5.55)         | 46.37 (-23.89)         | 64.14 (-1.10)         |
> Qwen2-VL-72B-3example   | 61.61 (+5.08)         | 69.46 (-6.93)         | 48.87 (-21.39)         | 63.13 (-2.11)         |
> InternVL2-40B           | 64.57                 | 74.12                 | 67.14                  | 67.95                 |
> InternVL2-40B-1example  | 54.01 (-10.56)        | 66.62 (-7.50)         | 36.97 (-30.17)         | 56.82 (-11.13)        |
> InternVL2-40B-2example  | 52.37 (-12.20)        | 65.24 (-8.88)         | 36.24 (-30.90)         | 55.37 (-12.58)        |
> InternVL2-40B-3example  | 51.05 (-13.52)        | 63.73 (-10.39)        | 39.94 (-27.20)         | 54.51 (-13.44)        |
>
>
> Q2: We have computed the proportion of multi-hop QA pairs in our benchmark, which is **2,459 out of 4,342, amounting to 56.63%**. We compared the models' performance on multi-hop versus non-multi-hop questions, and the results are shown in the table. From these results, we observe that models struggle with multi-hop reasoning tasks. We also provide examples of the multi-hop questions in the revised paper; please see Figure 16.
>
> Model                        | Perception | Reasoning | Probing | Overall |
> |-|-|-|-|-|
> InternVL2-40B-non-multi-hop       | 74.11          | 66.52         | -           | 72.28       |
> InternVL2-40B-multi-hop           | 51.24          | 77.01         | 59.59       | 64.63       |
> Qwen2-VL-72B-non-multi-hop        | 55.05          | 69.37         | -           | 58.55       |
> Qwen2-VL-72B-multi-hop            | 58.91          | 79.22         | 69.57       | 70.22       |
> VILA-40B-non-multi-hop            | 66.29          | 61.49         | -           | 65.14       |
> VILA-40B-multi-hop                | 44.58          | 71.62         | 62.16       | 60.25       |
> GPT-4o-non-multi-hop              | 63.19          | 57.77         | -           | 61.90       |
> GPT-4o-multi-hop                  | 48.51          | 66.76         | 54.65       | 58.03       |
> LLaVA-1.6-34B-non-multi-hop       | 66.14          | 61.27         | -           | 64.98       |
> LLaVA-1.6-34B-multi-hop           | 44.20          | 57.91         | 58.17       | 53.09       |
> Gemini-1.5-Pro-non-multi-hop      | 55.68          | 46.61         | -           | 53.50       |
> Gemini-1.5-Pro-multi-hop          | 42.39          | 62.78         | 49.60       | 53.09       |
>
>
> Q3: We have addressed the questions in points 1 and 2, the results are shown in the corresponding tables.

---

> > ### Comment · Reviewer_SeFC · 2024-11-21
> >
> > Regarding Q1: Could you clarify your In-Context Learning (ICL) setup? It’s a bit unclear whether the examples you use are specifically chosen to improve the final results (as they should be) or are selected randomly. Could you provide an example of an ICL configuration you’re using? For inspiration, you could refer to the Seed benchmark, which includes an ICL test. While it doesn’t explicitly target compositionality, it does assess it indirectly.
> >
> > For Q2: Thank you! In Table 11, I noticed that degradation occurs only in perception for multi-hop questions, while reasoning remains unaffected. This seems counterintuitive, as multi-hop questions are inherently more challenging and should impact reasoning. Could you revisit this? Why do you think this happens? Could it be an inherent issue in the data creation process? For your multi-hop examples, it might be helpful to provide an example for each topic to better illustrate the setup.

---

> > > ### Author Response · Authors · 2024-11-22
> > >
> > > Thank you for your thoughtful feedback! We appreciate the opportunity to analyze the underlying reasons behind the observed phenomena.
> > >
> > > Q1: In the initial ICL setting, we randomly sampled examples from the dataset to serve as in-context examples, testing with one, two, and three examples. We observed that increasing the number of examples led to a performance decline. We hypothesize that this decrease is due to the significant increase in prompt length caused by the additional examples, which negatively impacts the model's performance. Additionally, the in-context examples might confuse the models, leading to incorrect predictions.
> > >
> > > In this response, we employed a dynamic retrieval method that retrieves the most similar QA pairs for each query as in-context examples using Dense Passage Retriever(DPR) [2]. Additionally, we experimented with modified prompt formats inspired by SeedBench, integrating these structured prompts with the retrieved QA pairs. The results of these experiments are presented in the accompanying table, where we find that the dynamic retrieval method improves the model's performance on perception tasks compared to randomly sampled examples.
> > >
> > > Regarding the probing task performance in this response, since all the questions are indefinite choice, we hypothesize that including context examples may confuse the models about the number of correct answers. This could potentially lead to a decrease in performance, as the models might struggle to determine how many options to select based on the examples provided.
> > >
> > >
> > > In conclusion, optimizing ICL settings to maximize model performance requires significant effort and careful exploration. Given the complexity and variability of factors influencing ICL effectiveness, we believe it is valuable to leave further investigations and refinements to future work, providing ample space for continued exploration in this area.
> > >
> > > Model                | Perception         | Reasoning          | Probing             | Overall            |
> > > |-|-|-|-|-|
> > > Qwen2-VL-72B            | 56.53                 | 76.39                 | 70.26                  | 65.24                 |
> > > Qwen2-VL-72B-1example-short-prompt   | 62.07 (+5.54)         | 71.60 (-4.79)         | 42.41 (-27.85)         | 63.48 (-1.76)         |
> > > Qwen2-VL-72B-1-random-example (SeedBench Prompt Format)   | 62.05 (+5.52)         | 70.59 (-5.80)         | 49.67 (-20.59)         | 63.87(-1.37)         |
> > > Qwen2-VL-72B-1-DPR-retrieval-example (SeedBench Prompt Format)    | 62.69 (+6.16)         | 68.70 (-7.69)         | 46.48 (-23.78)         | 63.17 (-2.11)         |
> > >
> > >
> > > Q2: The observed results can be attributed to differences in difficulty distribution between the settings. As shown in the table, multi-hop perception contains a higher percentage of hard and super hard questions, while multi-hop reasoning includes a larger proportion of easy questions. Specifically, 38.64% of multi-hop reasoning questions are classified as easy, compared to only 3.72% in non-multi-hop reasoning. Therefore, the overall difficulty of multi-hop reasoning is lower than that of non-multi-hop reasoning. We believe this explains the phenomenon. **In addition, we have updated Figure 16 to include the 13 categories of multi-hop questions for enhanced clarity and comprehensiveness.**
> > >
> > > | Question       | Task | Easy      | Medium   | Hard     | Superhard|All       |
> > > |-|-|-|-|-|-|-|
> > > | Multi-hop    | Reasoning  | 437 (38.64%)   | 245 (21.66%)  | 399 (35.28%)   | 50 (4.42%)     | 1,131          |
> > > |                    | Perception | 3 (0.36%)      | 276 (33.29%)  | 373 (44.99%)   | 177 (21.35%)   | 829            |
> > > | Non-multi-hop  | Reasoning  | 17 (3.72%)     | 142 (31.07%)  | 212 (46.39%)   | 86 (18.82%)    | 457            |
> > > |                    | Perception | 85 (6.15%)     | 322 (23.28%)  | 756 (54.66%)   | 220 (15.91%)   | 1,383          |
> > >
> > > [2] Karpukhin, Vladimir, et al. "Dense passage retrieval for open-domain question answering." arXiv preprint arXiv:2004.04906 (2020).

---

> > > > ### Comment · Reviewer_SeFC · 2024-11-24
> > > >
> > > > I sincerely appreciate your hard work and effort. However, it appears that the multi-hop questions, which are a key component, were not executed well, particularly when the reasoning is relatively easy. Additionally, creating an in-context test set requires careful thought and attention to detail. That said, I will keep my current score as a reflection of my respect for your dedication.

---

### Official Review · Reviewer_wuw2 · 2024-11-01

**Soundness:** 2
**Presentation:** 2
**Contribution:** 2
**Rating:** 5
**Confidence:** 4

**Summary:**

The paper proposes a new compositional reasoning benchmark that is constructed from existing benchmarks (data collection, lines 211-238) augmented with negative options retrieval by similarity, consensus filtering by several recent LMMs and further human filtering of the resulting visual QA. An extensive evaluation of recent models on the proposed benchmark is performed. Some additional ablations are attempted by grouping models trained on more data, larger LLM decoders, vis. encoder combinations etc. However, those only confirm known facts: larger data, larger decoders, or more encoders are beneficial. Some analysis of failures is provided, albeit only qualitative. Main interesting aspect seems to be a large gap reported between human performance and the models. However, no statistics of the human subjects are provided (eg how many humans were employed, how they were motivated, what was the disagreement between humans, age groups, etc.).

**Strengths:**

* new benchmarks are always good, human curation is appreciated
* large number of models evaluated

**Weaknesses:**

* no new data is introduced, built from existing benchmarks
* no surprising conclusions
* no statistics for the human subjects
* error analysis is only qualitative
* dataset construction methodology involving humans could be more interesting - eg. humans could generate questions, red-team models to generate hard negatives etc.
* I disagree with Table 1, many benchmarks, including those listed have fine-grained questions, there are benchmarks (eg NVLRv2) involving multiple images, other benchmarks have human filtering, at least a partial subset, the only thing I indeed did not encounter before is "multiple right answers" (indefinite choice) - which could indeed be a contribution of the paper
* while benchmark contributions are appreciated, it seems this paper is somewhat below what I would expect from the level of contribution of an ICLR paper

**Questions:**

please see weaknesses

**Details Of Ethics Concerns:**

humans involved in data curation, without reporting details on that, however I am not sure if this is a real ethical concern here

---

> ### Author Response · Authors · 2024-11-20
>
> We sincerely appreciate your invaluable feedback and the opportunity to address your queries regarding our benchmark.
>
> Q1: As described in lines 132–133 of our paper, our proposed benchmark is a new human-annotated dataset for evaluating VLMs’ compositionally. **All the question-answer pairs are human-annotated (see lines 270–272).** We use various datasets with the potential to construct VL compositional QA pairs as our seed data (lines 212–213). Importantly, **we only use the images from these seed datasets and their initial annotations as prompts to construct new QA pairs**. Therefore, we have introduced new data in this work.
>
> Q2: The goal of our work is to provide a comprehensive diagnostic analysis of current VLMs regarding their capability for VL compositional perception and reasoning, serving as a complement to earlier comprehensive benchmarks such as MMBench, MMStar, and MME. Therefore, the significance of our work should not be overlooked. We believe that our contribution is non-trivial and will benefit the community involved in VLM design and training.
>
> Q3: We have updated the human subjects in the revised paper, please see Table 2 and Reviewer 33Vo Q3.
>
> Q4: We provide the quantitative results of our error analysis in the appendix; please see Figure 13 and Section A.4.
>
> Q5: As described in lines 132–133 of our paper, our dataset is **fully human-annotated**. **All the QA pairs are human-annotated (lines 270-272)**.  Therefore, the problem you mentioned is not apply for our paper.
>
> Q6: Please refer to Q1 and Q2, we believe our contribution is non trivial and will benefit the community of VLM design and training.
>
>
> In conclusion, we introduce a new dataset and all the QA pairs in our data are created by human annotators, as  **clarified in multiple sections of our paper**. We have carefully verified the properties that distinguish our dataset from previous work. We believe that our contribution is significant and will benefit the VLM community.

---

> > ### Comment · Reviewer_wuw2 · 2024-11-25
> > **rebuttal response**
> >
> > thank you for your response.
> > 1. by no "new data" I was referring to basing on existing datasets for the image and initial metadata source
> > 2. there are a bunch of compositional reasoning benchmarks proposed in the past - aro, crepe/sugarcrepe, vl-chcklist, eyes wide shut, and newer ones more recently pushed to arxiv - all of them offer some data collection methodology, most if not all have human filtering partitions. Your work is certainly valuable, yet it would be nice to see some more clear distinction - eg something specific that is only detectable by your benchmark
> > 3. Having a resolution gap analysis (higher then 768 vs lower then 768) and verifying average score of 54 models is below average is a good start (btw, 54 models' average lower than chance sounds suspicious as it might be biased towards the weak models, how about the same for strongest-k models? ideally maybe k=1 or k=2?), yet I would expect more in-depth insights from a benchmark paper - detailed analysis on what is difficult to the strongest models in terms of different expected human capabilities, comparing those individually to human performance etc. So please don't be discouraged by my comment, all I am saying is that with more work I believe your efforts would indeed make this a significant tool for the community, I just feel it is not quite there yet.
> > 4. "will benefit the community of VLM design and training" - for this to be so, you need to pinpoint more clearly what is that your benchmark currently predicts the community needs to focus on? for example higher resolution handling cannot be it, as this was already quite extensively popularized by the llava team with their extended anyres experiments (pls see one of their blogs), etc
> >
> > In light of the above reasoning, I still prefer to keep my current score of 5, but encourage the authors to continue working on their benchmark and submit it to a later venue, I just don't think it is yet ready in its current state

---

> ### Author Response · Authors · 2024-11-25
>
> Thank you for your feedback. We are pleased to address your remaining concerns.
>
> Q1. There are no restrictions against using existing data as seed data for dataset construction. Many new datasets, such as RefCoCo, Visual Genome, LLaVA Bench, and MMVP, leverage existing data sources in this way and are well-accepted as significant tools for VLM evaluation and diagnosis. Additionally, you recognize our dataset as "a new benchmark" in the Strengths section. We believe this point is unrelated to the core contributions of our paper and seems to focus excessively on minor or irrelevant details.
>
>
> Q2. We have included the main differences between our work and existing benchmarks. **Table 1 and Section 1 clearly outline the novelty and distinct aspects of MMComposition compared to existing works.**  Reviewer 1 has recognized the novelty and distinct aspects of our work, please also refer to R1 Q1, where we have successfully addressed this concern for them.
>
> Q3. The resolution gap is a significant factor that may influence a model's capabilities. It is reasonable that models limited to processing low-resolution images exhibit poorer compositionality. Therefore, your statement that "it might be biased towards the weak models" is consistent with intuition. We have analyzed the impact of encoding resolution on models' performance in Section A.2 for the top three models (k=3). **Thus, your concern regarding ‘the same for strongest-k models? ideally maybe k=1 or k=2’ has been addressed in our paper.** In addition, our paper includes an in-depth analysis of the strongest models in terms of different expected human capabilities—these models include GPT-4o, Qwen2-VL, and InternVL. Furthermore, our work provides an in-depth analysis of the relationship between the design of VLMs and their compositionality. As the first study to comprehensively examine complex compositional perception and reasoning, it distinguishes itself from previous works such as ARO, Crepe, and SugarCrepe. **We want to emphasize that this contribution should not be overlooked.**
>
> Q4. Our paper clearly "predicts the community needs to focus on". In Section 1 (lines 106–126), we have explicitly addressed this by highlighting: **"(1) Visual Encoder Design: While a mixture-of-encoder architecture can enhance compositionality, adding more encoders does not necessarily improve performance." ... "we find that for relatively simple QA tasks, only a small portion of its language capabilities are utilized ... Once the language decoder size reaches a certain threshold (e.g., 34B, 70B), the visual encoder has a more significant impact on the model’s compositionality. ")**. We believe these points provide clear guidance on where the community's efforts could be most effectively directed.
>
> In contrast, the LlaVA team's analysis **primarily identifies the phenomenon but does not delve into the underlying reasons. Furthermore, prior works lack a comprehensive analysis of why models fail at fine-grained visual compositional perception and reasoning. In our paper, we thoroughly examine these underlying reasons in Sections 1, 5, and A.2.** Therefore, we believe this concern has already been addressed in our work.
>
>
> In conclusion, we believe that conclusions should be based on **evidence**, not merely on **” reasoning.”** The concerns you mentioned have been thoroughly addressed in our work, and some of the "issues" you raised have, in fact, been recognized as strengths by other reviewers. Therefore, we respectfully request a reconsideration of the evaluation of our paper.

---

> ### Author Response · Authors · 2024-11-27
>
> Dear Reviewer wuw2,
>
> We believe we have addressed your concerns.  If our careful and respectful  responses continue to be ignored, we will report this to the ACs and/or PC.

---

### Official Review · Reviewer_33Vo · 2024-11-03

**Soundness:** 2
**Presentation:** 2
**Contribution:** 2
**Rating:** 6
**Confidence:** 4

**Summary:**

This paper introduces MMComposition, a QA benchmark that evaluates the compositional capabilities of modern vision-language models. MMComposition encompasses a range of tasks, including perception, reasoning, and probing, with multiple subtasks presented in various QA formats: yes/no, multiple-choice, and indefinite-choice. The dataset is curated from numerous existing sources, with QA pairs annotated by humans. Covering 13 distinct vision-language compositionality tasks, this benchmark offers a comprehensive evaluation of both proprietary and open-source vision-language models. The paper also analyzes factors that may influence the compositional abilities of VLMs, such as the resolution of visual encoders, the scale of language decoders, and the volume of training data.

**Strengths:**

- This paper presents a comprehensive benchmark focused on compositionality, encompassing a wide range of skills from perception and reasoning to probing.

- This paper provides an extensive evaluation of recent models, including both open-source and API-based models, highlighting areas where they continue to fall short of human capabilities.

- The paper is well-written with clearly organized sections.

**Weaknesses:**

- Although the benchmark includes diverse skill sets and QA formats, the specific aspects that pose challenges are not clearly defined. It is also unclear what distinguishes this benchmark from other general QA datasets designed to test modern VLMs for AGI, such as MMMU, MMStar, and tons of similar benchmarks. The paper does not provide comparisons in terms of general capabilities across QA datasets; instead, it focuses on embedding-based benchmarks for comparison, as shown in Table 1. Comparing the scale of evaluation samples, such as the number of images or questions across different benchmarks, would also be valuable.


- Related to the first weakness, one might question whether this benchmark is truly challenging. Some compositionality benchmarks or visual QA tasks could potentially be solved using only language models in an image-blind setting, due to language priors, such as coherence, grammar, and clues embedded across answer choices. As specific example, in second example in figure 3, can is often made of metal, such knowledge aids in answering correctly without relying on visual cues. It would be beneficial to examine the proportion of questions that can be solved solely using large language models.


- Several essential details are missing regarding the benchmark construction. In the human annotation process, additional information is needed: Who annotated the dataset? How was confidence measured, and how were errors handled in finalizing the annotations? Additionally, it’s unclear how misaligned captions were manually added in the probing task (line 255). Furthermore, for reporting human performance, what was the process? It would be important to present individual human performance scores for each skill, rather than a single overall score.


- The empirical trends concerning the scale of the visual encoder, language decoder, and training data are perhaps not surprising. The paper does not analyze whether these trends are specific to the proposed benchmark or if they also appear in other general visual QA benchmarks. Meanwhile, an additional suggested analysis could explore how the design of the visual connector (e.g., fully connected layer or Q-Former style) and the method of visual token insertion (e.g., tokens input directly into the language model or through cross-attention connections) impact performance of the proposed benchmark.


- There are some notable clarity issues, including typographical errors such as 'MuriBench' in line 237 and 'ARC' in line 241. Additionally, there are inconsistencies in publication years for certain cited papers, particularly recent NeurIPS papers like SugarCrepe, which collectively raise concerns about professionalism.


- Could fine-tuning VLMs on specific datasets improve performance on MMComposition?

---

Assessment: While the extensive evaluations across VLMs are commendable, the benchmark falls short of expected standards in terms of detailed documentation, verification, and comparisons with other QA benchmarks. Additionally, analyses of the proposed benchmark could be enhanced by comparing observed trends with those from other benchmarks.

**Questions:**

- The reasoning behind the name 'MMComposition' is unclear.

---

> ### Author Response · Authors · 2024-11-20
> **Part 1**
>
> Thank you for the time, thorough comments, and nice suggestions. We hope our response can adequately address your concerns.
>
> Q1: We have included a comparison of our benchmark with other general benchmarks, as shown in the table. Our benchmark stands out from well-known benchmarks due to its multi-hop QA pairs, its specific capability assessment -- compositionality -- and its challenging nature. This comparison has also been included in the revised paper (see Table 8).
>
> Dataset | Size | Human Annotation | Multi-Hop | Capabilities | Best Performance (Model/Human) |
> |-|-|-|-|-|-|
> MMBench | 3,217    | ✗                     | ✗             | Comprehensive                             | 86.1 / -                           |
> MME  | 2,800    | ✓                     | ✗             | Comprehensive                             | 1790.04 / -                        |
> MMStar | 1,500    | ✓                     | ✗             | Comprehensive                             | 66.0 / -                           |
> SeedBench | 19k     | ✓                     | ✗             | Comprehensive                             | 72.4 / -                           |
> MMMU | 11.5k    | ✓                     | ✗             | College-Level Subject Knowledge          | 69.1 / 88.6                        |
> HalBench | 1,129    | ✓                     | ✗             | Hallucination                             | 67.58 / -       |
> **MMComposition (ours)** | 4,342    | ✓                     | ✓             | **Compositionality**  | 67.95 / 90.31      |
>
> Q2: To verify the challenging nature of our dataset and demonstrate the indispensable role of images, we conducted experiments comparing the models' performance between the standard setting and an image-blind setting. As shown in the table below, without image input, the models' performance decreases significantly, indicating that they must rely on image compositional information to obtain the correct answers. This result has also been included in the revised paper (see Table 10).
> Image-Blind Setting:
> |Model|Perception|Reasoning|Probing|Overall|
> |-|-|-|-|-|
> |Qwen2-VL-72B|56.53|76.39|70.26|65.24|
> |Qwen2-VL-72B-blind|45.16 (-11.37)|48.17 (-28.22)|30.76 (-39.50)|44.74 (-20.50)|
> |InternVL2-26B|60.40|70.03|52.43|63.08|
> |InternVL2-26B-blind|34.80 (−25.60)|42.63 (−27.40)|32.17 (−20.26)|37.39 (−25.69)|
> |InternVL2-40B|64.57|74.12|67.14|67.95|
> |InternVL2-40B-blind|37.88 (−26.69)|43.35 (−30.77 )|34.28 (−32.86)|39.54 (−28.41)|
> |InternVL2-76B|63.41|75.44|58.46|67.28|
> |InternVL2-76B-blind|33.93 (−29.48)|44.08 (−31.36)|32.68 (−25.78 )|37.51 (−29.77)|
>
> Q3: The data was initially annotated by student workers and then verified by another group of workers; finally, the dataset was refined and finalized by the authors. We followed the method proposed in FineMatch [1], which involves replacing attribute, relation, and object phrases while maintaining the Part of Speech (POS) tags unchanged. This approach ensures that mismatched captions remain as similar as possible at the character level to the original correct captions. We have updated the human performance metrics for each task in **Table 2 of the revised paper**.
> [1] "FineMatch: Aspect-Based Fine-Grained Image and Text Mismatch Detection and Correction." ECCV, 2024.
>
> Q4: We have clarified in our paper (lines 122–125) that the visual encoder plays a more significant role in the compositionality of VLMs. Models with enhanced capabilities for perceiving fine-grained compositional image information can provide more detailed inputs to language models. Moreover, we have added a comparison of different visual connectors, including Q-Formers and MLP models, with the results shown in the Table below. From these results, we conclude that the Q-Former architecture cannot provide detailed visual references to language models for fine-grained compositional image understanding. This result has also been included in the revised paper (see Table 9).
> Model | Visual Encoder | LLM | V2L Adapter | Perception | Reasoning | Probing | Overall |
> |-|-|-|-|-|-|-|-|
> mPLUG-Owl2                       | ViT-L/14           | LLaMA2-7B      | Q-Former        | 36.90          | 46.16         | 30.36       | 39.59       |
> InstructBLIP-7B                  | ViT-G/14           | Vicuna-7B      | Q-Former        | 33.22          | 43.70         | 31.41       | 36.86       |
> LLaVA1.5-7B                         | ViT-L/14           | Vicuna-7B      | MLP             | 36.51          | 47.04         | 30.32       | 39.71       |
> InstructBLIP-13B                 | ViT-G/14           | Vicuna-13B     | Q-Former        | 35.53          | 42.70         | 25.24       | 37.06       |
> LLaVA1.5-13B                        | ViT-L/14           | Vicuna-13B     | MLP             | 37.23          | 49.75         | 39.32       | 42.03       |
>
> Q5: Thank you for highlighting this issue. We have addressed it in the revised version.

---

> ### Author Response · Authors · 2024-11-20
> **Part 2**
>
> Q6: Good point. We provide experimental results comparing models before and after tuning with extra instruction tuning data. Please refer to the Table below. The results indicate that models benefit from fine-tuning with extra related datasets for their compositional perception capability, and we have updated this finding to the revised version. In addition, Table 5 in our paper also compares the performance of the models with and without more data fine-tuning.
> Model | Perception | Reasoning | Probing | Overall |
> |-|-|-|-|-|
> LLaVA1.5-7B|36.51| 47.04                | 30.32               | 39.71               |
> LLaVA1.5-7B+ShareGPT4V     | **38.00**| 45.34| 26.43| **39.46**|
> LLaVA1.5-13B       | 37.23                | 49.75                | 39.32               | 42.03|
> LLaVA1.5-13B +ShareGPT4V    | **40.04**| 47.73| 39.29| **42.77**|
> *Comparison of LLaVA1.5 and LLaVA1.5 fine-tuned with the ShareGPT4V dataset in MMComposition.
>
> In conclusion,  our work aims to provide a comprehensive diagnostic analysis of current VLMs regarding their capability for VL compositional perception and reasoning, serving as a complement to earlier comprehensive benchmarks such as MMBench, MMStar, and MME.

---

> ### Comment · Reviewer_33Vo · 2024-11-22
>
> From the authors' response, the work appears to be improved, particularly in its comparison with previous QA benchmarks, its analysis of the benchmark's challenges (e.g., the image-blind setting), and the additional exploration of connector design and fine-tuning approach.
>
> Regarding the image-blind setting, I am curious whether pure language-based LLMs, such as LLaMA-3.1, GPT-3.5, or others, can perform well on the proposed benchmark in the absence of the provided image.

---

> ### Author Response · Authors · 2024-11-23
>
> Thank you for your endorsement and valuable suggestions! In response, we have included results for pure language models, specifically GPT-3.5, Qwen2.5-72B-Instruct, and LLaMA-3.1-70B. Please refer to the tables for detailed comparisons and insights. From the tables, we observe that the performance of the pure language models is close to random guessing (30.15%), which underscores the indispensable role of visual information in our dataset.
>
> Model                | Perception         | Reasoning          | Probing             | Overall            |
> |-|-|-|-|-|
> GPT-3.5-Turbo     | 26.53                 | 42.07                | 32.93                 | 32.89                |
> LLaMA-3.1-70B   | 36.15         | 35.08         | 26.58        | 34.74         |
> Qwen2.5-72B   | 37.16         | 40.49         | 30.76         | 37.70        |

---

> ### Comment · Reviewer_33Vo · 2024-11-24
>
> My initial concerns have been addressed. I would be happy to adjust my score recommendation to 6 if the evaluation code release for MMComposition includes procedures for testing the models and settings reported in the paper, including image-blind settings with both VLMs and LLMs.

---

> ### Author Response · Authors · 2024-11-24
>
> Thank you for your thoughtful feedback and for considering adjusting the score recommendation. We are pleased that our response has successfully addressed your initial concerns, and your suggestions have significantly helped us improve the quality of our work!
>
> We want to confirm that the example evaluation code for MMComposition is now complete and has been made available through the supplementary materials. This code provides comprehensive procedures for testing the models and settings discussed in our work, including the image-blind settings for both VLMs and LLMs. We believe this fully addresses your concerns regarding reproducibility and transparency. Furthermore, we are actively working on refining and formatting all evaluation codes to support a more comprehensive and robust evaluation framework for MMcomposition. All the resources will be released soon.
>
> Thank you again for your valuable time and thoughtful review!

---

> ### Comment · Reviewer_33Vo · 2024-11-25
>
> Thank you for the comment! I have updated the score accordingly.

---

### Author Response · Authors · 2024-12-03
**General Response to Reviewers and ACs**

We sincerely thank the reviewers for their thoughtful evaluations and constructive feedback. We are encouraged by the recognition of the strengths of our work, including:

- **Key Contributions**:
  - **R3**: *"MMCOMPOSITION evaluates tasks like multi-image reasoning, object interactions, and counting, all of which are crucial for real-world, nuanced understanding."*
  - **R3**: *"Improvement upon Existing Compositional Datasets."*
  - **R4**: *"The analysis on model component provides valuable insight on model design."*

- **Benchmark Design**:
  - **R1**: *"A comprehensive benchmark focused on compositionality, encompassing a wide range of skills."*
  - **R2**: *"New benchmarks are always good, human curation is appreciated."*
  - **R4**: *"Human-annotated and covers a wide range of tasks in terms of compositional understanding."*
  - **R4**: *"The benchmark is challenging and demonstrates a large performance gap between human and VLMs."*

- **Experiments and Analysis**:
  - **R1**: *"This paper provides an extensive evaluation of recent models."*
  - **R2**: *"Large number of models evaluated."*
  - **R3**: *"In-depth model comparison and component analysis."*

- **Paper Writing and Organization**:
  - **R1**: *"The paper is well-written with clearly organized sections."*

Meanwhile, we would like to raise some concerns regarding the comments from **R2** and **R3**:

- **R2 (wuw2)**:
  - It appears that R2 may not have thoroughly read and understood our paper, and our careful and respectful responses are deliberately ignored.
  - They missed most of the key points, and the concerns raised by R2 have been addressed in our paper revision.
  - Moreover, R2 made conclusions based on their "reasoning" rather than evidence. We believe that conclusions should be based on evidence, not merely on reasoning.

- **R3 (SeFC)**:
  - The final comments from R3 are ambiguous and fail to specify concrete issues with our work.
  - The concerns raised seem unrelated to the core contributions of our paper, instead focusing disproportionately on minor or peripheral details.
  - Despite multiple requests for clarification, we have not received any further response. This lack of engagement limits our ability to address their concerns effectively.

We believe we have thoroughly addressed all the main concerns that were clearly articulated.  It would be unfair to disregard our contributions based on these misunderstandings and ambiguous feedback. Therefore, we respectfully request that the Area Chair investigate these issues to ensure a fair evaluation.

Thank you again for the valuable dedication and for recognizing the significance of our contribution.

---

### Note · Authors · 2025-01-23

I have read and agree with the venue's withdrawal policy on behalf of myself and my co-authors.